# Neptune-X: Active X-to-Maritime Generation for Universal Maritime Object Detection

**Yu Guo**[1,3]**, Shengfeng He**[2,*]**, Yuxu Lu**[4]**, Haonan An**[1]**, Yihang Tao**[1]**,**
**Huilin Zhu**[5]**, Jingxian Liu**[3]**, Yuguang Fang**[1]

[1]Hong Kong JC STEM Lab of Smart City and Department of Computer Science,
City University of Hong Kong [2]Singapore Management University
[3]State Key Laboratory of Maritime Technology and Safety, Wuhan University of Technology
[4]The Hong Kong Polytechnic University [5]Wuhan University of Science and Technology
`https://github.com/gy65896/Neptune-X`

## Abstract

Maritime object detection is essential for navigation safety, surveillance, and autonomous operations, yet constrained by two key challenges: the scarcity of annotated maritime data and poor generalization across various maritime attributes (e.g., object category, viewpoint, location, and imaging environment). To address these challenges, we propose Neptune-X, a data-centric generative-selection framework that enhances training effectiveness by leveraging synthetic data generation with task-aware sample selection. From the generation perspective, we develop X-to-Maritime, a multi-modality-conditioned generative model that synthesizes diverse and realistic maritime scenes. A key component is the Bidirectional Object-Water Attention module, which captures boundary interactions between objects and their aquatic surroundings to improve visual fidelity. To further improve downstream tasking performance, we propose Attribute-correlated Active Sampling, which dynamically selects synthetic samples based on their task relevance. To support robust benchmarking, we construct the Maritime Generation Dataset, the first dataset tailored for generative maritime learning, encompassing a wide range of semantic conditions. Extensive experiments demonstrate that our approach sets a new benchmark in maritime scene synthesis, significantly improving detection accuracy, particularly in challenging and previously underrepresented settings.

## 1 Introduction

Object detection is a fundamental technology for maritime environmental perception, enabling the identification of object categories and the localization of bounding boxes in images captured by imaging systems deployed on various facilities, such as surface vessels, coastal infrastructure, and aerial platforms. It plays a key role in a variety of maritime applications, including autonomous or assisted navigation for surface ships [43], intelligent video surveillance for coastal facilities [40], and autonomous inspection using Unmanned Aerial Vehicles (UAVs) [44].

Despite rapid advances in deep learning-based object detection [39, 37, 23], the generalization capability of these models remains heavily dependent on the scale and diversity of annotated training data. Specifically, maritime-specific object detection datasets face two core limitations. *First*, the acquisition and annotation process is costly and labor-intensive. Different from land scenarios, data collection from heterogeneous platforms such as ships, UAVs, and stationary coastal cameras requires significant operational resources. In addition, the manual annotation of bounding boxes limits dataset scalability. *Second*, existing datasets exhibit large disparities in training difficulty across multiple

---

[*]Corresponding author: shengfenghe@smu.edu.sg.

39th Conference on Neural Information Processing Systems (NeurIPS 2025).

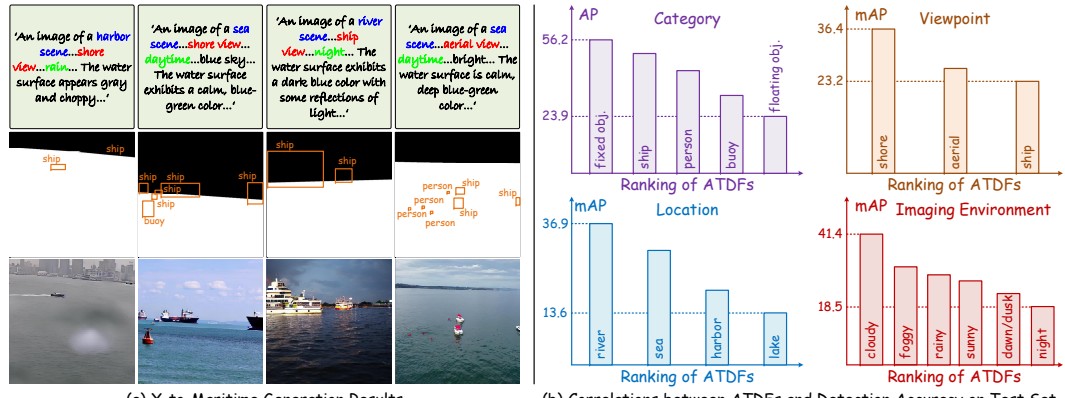

(a) X-to-Maritime Generation Results

(b) Correlations between ATDFs and Detection Accuracy on Test Set

Figure 1: We introduce Neptune-X, a generation-selection framework for robust maritime object detection. (a) It enables the first multi-modality-conditioned data generation, supporting diverse and controllable maritime scene synthesis. (b) Our active selection strategy leverages the Attribute-correlated Training Difficulty Factor (ATDF), which correlates with detection performance and guides the selection of high-value synthetic samples to optimize downstream detector accuracy.

attributes, including imaging conditions, viewpoints, water environments, and object categories. These imbalances are driven by uneven sample distributions and systematic biases in data collection, resulting in poor generalization to rare or complex maritime scenarios.

To mitigate data limitations, traditional data augmentation techniques [51, 48, 47, 2] have been widely adopted. These methods apply geometric transformations, color perturbations, and sample mixing to increase training diversity. However, they operate only on existing samples and cannot generate fundamentally new instances with novel semantics. As a result, they are insufficient for addressing performance degradation caused by data scarcity and semantic imbalance.

Compared to GANs [10, 8, 49, 20], the diffusion models [30] achieve superior image quality and training stability for more flexible text-conditioned image synthesis. Layout-to-Image techniques [21, 46, 41] further enhance controllability by conditioning on layout information, such as bounding boxes, and have been explored to synthesize detection-specific training data [34]. However, applying these methods directly to the maritime domain remains problematic. Maritime scenes demand an explicit semantic understanding of the interaction between objects and their aquatic surroundings, as objects and environmental context (e.g., sea states, reflections, lighting) are closely intertwined in both appearance and meaning. Without modeling this relationship, generative models often produce semantically inconsistent and physically implausible artifacts, such as ships floating unnaturally in mid-air or disconnected from the water surface. Furthermore, existing approaches overlook the fact that synthetic samples vary in training utility due to differences in category, viewpoint, and condition. Failing to account for such disparities leads to suboptimal data selection and limited gains.

In this paper, we introduce Neptune-X[2], a unified data generation-selection paradigm designed to address the dual challenges of data scarcity and limited diversity in maritime object detection. Our approach fuses multi-condition maritime scene generation with task-aware sample selection to enhance both quantity and quality of training data. In the generation phase, we develop a controllable generative framework that supports diverse input modalities and produces semantically rich maritime scenes. A central innovation is the Bidirectional Object-Water Attention (BiOW-Attn) mechanism, which explicitly models interactions between objects and their aquatic surroundings to improve the realism and coherence of object placement. This enables the generation of visually plausible maritime scenes with fine-grained spatial semantics (Fig. 1a).

For data selection, Neptune-X incorporates an Attribute-dependent Active Sampling (AAS) strategy to prioritize training samples that are most beneficial for detection performance. This strategy is guided by Attribute-related Training Difficulty Factors (ATDF), which estimate the learning difficulty associated with different semantic attributes, such as viewpoint, object category, and environmental condition. As illustrated in Fig. 1b, ATDF captures the relative training complexity across attributes and informs the weighting of synthetic samples during selection. By aligning sample value with

---

[2]The model name "Neptune-X" integrates the marine symbolism of Neptune (Roman sea god) with multi-modality conditional guidance (X).

task difficulty, AAS enables more focused and efficient use of generated data, ultimately guiding the detector to learn from challenging and underrepresented cases.

To support training and evaluation under diverse maritime conditions, we construct a new benchmark, the Maritime Generation Dataset, which covers a broad range of scenarios with variations in object category, viewpoint, environment, and location. Extensive experiments demonstrate the effectiveness of Neptune-X in both synthetic scene quality and downstream detection performance, particularly in challenging and underrepresented cases.

In summary, our main contributions are threefold:

- We present the X-to-Maritime generation framework for maritime scenes, featuring a Bidirectional Object-Water Attention module that enhances realism by modeling object-water interactions under multi-condition inputs.

- We propose an Attribute-dependent Active Sampling strategy that estimates training difficulty across semantic dimensions and selects high-value samples through difficulty-aware weighting.

- We construct a Maritime Generation Dataset, the first generative benchmark for maritime detection. Experiments show that our method improves both generation quality and detection performance in challenging scenarios.

## 2   Related Work

**Diffusion-based Image Generation.** Diffusion models [14] have demonstrated strong cross-modal generation capabilities and have been widely adopted for image synthesis tasks. Early works such as DALL-E 2 [26] employed hierarchical diffusion guided by CLIP text encoders [25] to achieve text-to-image generation, while Imagen [31] further improves generation quality by leveraging the language understanding power of large-scale language models. Stable Diffusion (SD) [30] made this technology more accessible by introducing latent space compression, enabling efficient high-resolution synthesis. However, text-only conditioning often lacks fine-grained spatial and attribute-level control. To address this, recent layout-to-image (L2I) methods incorporate auxiliary conditions for more precise generation. For example, LayoutDiff [50] integrates bounding box constraints into the diffusion process. GLIGEN [18] introduces gated self-attention to fuse layout and textual conditions in a pre-trained SD model, while RC-L2I [5] employs regional cross-attention to enhance instance-level controllability. However, these methods generally treat object regions independently and overlook interactions with complex scene contexts, limiting their effectiveness in domains like maritime environments.

**Data Augmentation for Object Detection.** Early data augmentation strategies such as Mixup [48], CutMix [47], and Mosaic [2] primarily rely on pixel-level rearrangements to increase visual diversity. While effective in remixing existing patterns, these methods are limited in their ability to produce novel samples that extend beyond the original training distribution. This constraint has driven recent interest in generative augmentation, where image synthesis models are used to create new samples with richer semantics and structural variability. For instance, DA-Fusion [36] employs a diffusion model to enhance dataset diversity, while Fang et al. [11] introduce a visual prior-guided controllable diffusion framework for object detection. AeroGen [34] further explores layout-conditioned diffusion to generate synthetic remote sensing imagery based on rotated bounding boxes. However, most existing generative augmentation methods focus solely on generation and overlook the importance of evaluating the training utility of synthesized samples, which makes it difficult to prioritize data that maximally benefits downstream learning.

**Active Learning for Object Detection.** Active learning for object detection aims to boost model performance with minimal labeling effort by selecting the most informative unlabeled samples. Early methods [32, 45, 1] adapt classification-based strategies but often ignore the localization task. To address this, later works introduce uncertainty-based approaches. For example, Choi et al. [7] modeled joint uncertainties using Mixture Density Networks, while PPAL [42] combines difficulty-calibrated uncertainty sampling with diversity-based selection. In our setting, although synthetic images exhibit semantic diversity, their impact on detector training varies. To improve efficiency, we draw on active learning principles to design a sample selection strategy tailored for generated data. Unlike conventional methods, which rely on human annotation, our approach leverages known labels from the generation process, avoiding annotation costs while enabling difficulty-aware selection.

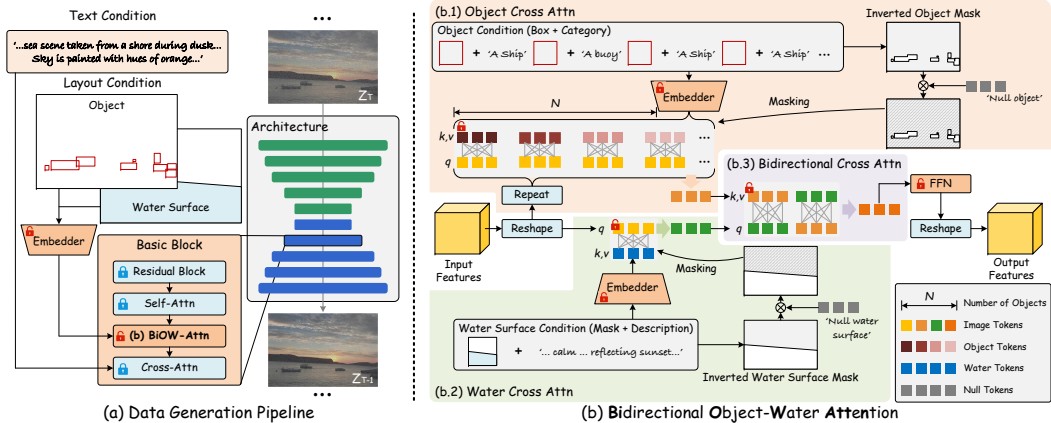

Figure 2: Architecture of our X-to-Maritime generator. BiOW-Attn serves as the core component for integrating object and water surface conditions.

# 3 Neptune-X

This section introduces two key components of the proposed Neptune-X, including X-to-Maritime Generation and High-quality Data Generation.

## 3.1 X-to-Maritime Generation

**Latent Diffusion Model.** As a state-of-the-art approach for text-to-image generation, SD [30] establishes itself as an effective framework for generative modeling. In this paper, we adopt SD as the foundational architecture due to its demonstrated effectiveness in conditional image generation. To be specific, SD is a classic Latent Diffusion Model (LDM), comprising two parts:

- **Latent Space Projection:** A Variational Auto-Encoder (VAE) learns bidirectional mappings between pixel-level images $I \in \mathbb{R}^{H \times W \times 3}$ in RGB space and compressed latent codes $z \in \mathbb{R}^{h \times w \times c}$, where $c$ denotes the number of channels, while $h = H/m$ and $w = W/m$ denote the reduced spatial dimensions through a compression factor $m$. This dimensionality reduction enables computationally efficient diffusion processes while preserving essential visual features.

- **Conditional Diffusion Process:** SD employs text-conditioned diffusion training through caption embeddings $\mathcal{C}$, implementing a Markov chain that gradually denoises latent representations across $T$ timesteps. The denoising operator $g_\theta$, parameterized by $\theta$, is optimized to estimate noise components through a noise prediction objective, which can be given by

$$\mathcal{L} = \mathbb{E}_{z,t,\epsilon \sim \mathcal{N}(\mathbf{0},\mathbf{I})} \left[ \|\epsilon - g_\theta(z_t, t, \mathcal{C})\|_2^2 \right], \tag{1}$$

where $z_t$ represents the noisy latent at timestep $t$. This formulation enables stable SD training through gradient updates while maintaining semantic alignment between the caption conditions and generated images.

**Multi-Condition Guidance.** Despite extensive training in the SD model and its remarkable efficacy in text-to-image synthesis, effectively incorporating layout conditions to jointly guide the maritime scenario generation process remains challenging. To address this limitation, we design a novel domain-specific model (named Neptune-X) for the generation of maritime images, as shown in Fig. 2a. In particular, we propose a well-designed Bidirectional Object-Water Attention (BiOW-Attn) module to integrate additional layout conditions from the water surface targets and the water body itself, thereby enhancing the generative capability and controllability in maritime scenarios. To enable multi-condition guidance, we thus extend the denoising objective in Eq. (1) as

$$\mathcal{L} = \mathbb{E}_{z,t,\epsilon \sim \mathcal{N}(\mathbf{0},\mathbf{I})} \left[ \|\epsilon - g_\theta(z_t, t, \mathcal{C}, \underbrace{\{\mathcal{C}_o^i, \mathcal{M}_o^i\}_{i=1}^O}_{\text{object conditions}}, \underbrace{\{\mathcal{C}_w, \mathcal{M}_w\}}_{\text{water surface condition}})\|_2^2 \right], \tag{2}$$

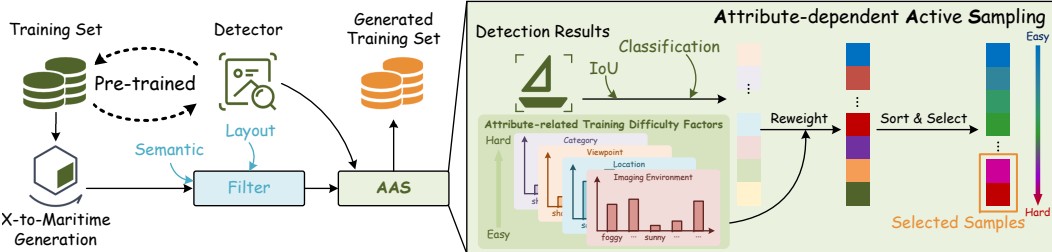

Figure 3: Data generation flowchart of Neptune-X. AAS holistically integrates both detection accuracy and training difficulty through introducing the attribute-related training difficulty factors as weights to select high-value samples generated by X-to-Maritime.

where $O$ denotes the number of total objects, $\mathcal{C}_o^i$ and $\mathcal{M}_o^i$ represent the $i$-th object's feature embedding and binary spatial mask, respectively, and $\mathcal{C}_w$ and $\mathcal{M}_w$ denote the water surface feature embedding and corresponding binary mask.

**Layout Condition Embedders.** Inspired by GLIGEN [18], our module employs an identical embedding strategy to transform both conditional types into token representations. For the $i$-th object with class label $L_o^i$ and spatial coordinates $P_o^i$, we first encode the coordinates through Fourier embedding $\Phi$ to obtain positional features $\mathbf{e}_o^i = \Phi(P_o^i)$. Simultaneously, the textual label $L_o^i$ is encoded by a CLIP text encoder $\xi$ into semantic tokens $\mathbf{t}_o^i = \xi(L_o^i)$. The final feature embedding of the $i$-th object $\mathcal{C}_o^i$ is then derived through channel-wise concatenation $[\,\cdot\,;\,\cdot\,]$ and MLP projection, i.e.,

$$\mathcal{C}_o^i = \text{MLP}\left(\left[\mathbf{e}_o^i; \mathbf{t}_o^i\right]\right). \tag{3}$$

Similarly, we obtain the water surface embedding $\mathcal{C}_w$ following the same procedure as object embedding, where we replace $L_o^i$ with a water surface description $L_w$ and $P_o^i$ with the minimum enclosing rectangle $P_w$ of the water mask.

**Bidirectional Object-Water Attention.** As shown in Fig. 2b, we propose BiOW-Attn to enable the targeted generation of maritime scenarios, which comprises two stages, i.e., conditional integration and bidirectional feature interaction. In the first stage, cross-attention modules independently process each object and water embeddings through identical operations applied to the input feature and each embedding, generating conditionally augmented features as defined mathematically by

$$\text{Cross-Att}(Q, K_k, V_k) = \text{Softmax}\left(\frac{Q \cdot K_k^\top}{\lambda}\right)V_k, \quad k \in \{\mathcal{C}_o^i, \mathcal{C}_w\}, \tag{4}$$

where $\lambda$ is a scaling factor. $Q$ denotes the query vector computed from input features, while $\{K_k, V_k\}$ corresponds to the key-value pairs projected from water and object embeddings.

For the object cross attention module in Fig. 2b.1, the enhanced features $\{f_o^i\}_{i=1}^O$ guided by each object are summed to produce the output. The aggregated output is then masked using the union of all object masks $\{\mathcal{M}_o^i\}_{i=1}^O$, while non-object regions are filled with a learnable null object embedding $\mathbf{null}_{\text{obj}}$ to stabilize spatial localization. Mathematically, the output $\mathbf{F}_o$ can be generated by

$$\mathbf{F}_o = \left(\sum_{i=1}^O f_o^i\right)\odot\mathbf{M} + \mathbf{null}_{\text{obj}}\odot(1-\mathbf{M}), \quad \text{where } \mathbf{M} = \bigcup_{i=1}^O \mathcal{M}_o^i. \tag{5}$$

In contrast, the water cross attention module in Fig. 2b.2 follows an identical architecture to obtain the output $\mathbf{F}_w$, with $\mathbf{M}$ replaced by the water-specific mask $\mathcal{M}_w$ and $\mathbf{null}_{\text{obj}}$ substituted by the water null embedding $\mathbf{null}_{\text{wat}}$.

The second stage processes $\mathbf{F}_o$ and $\mathbf{F}_w$ generated from the previous step through bidirectional cross attention in Fig. 2b.3 by exchanging the input sources of query vectors and key-value pairs. This module significantly improves the object-water boundary interaction of generated images, thereby generating more physically plausible water surface targets. The final output is produced by a Feed-Forward Network (FFN).

## 3.2 High-quality Data Generation

Fig. 3 shows the data generation process of Neptune-X. Building upon the X-to-Maritime generation model constructed in Sec. 3.1, we employ random transformations (including randomly sampling image and water surface descriptions, and resizing or flipping annotated bounding boxes, etc.) on the existing training set labels to enhance the diversity of generated data. Subsequently, we apply a filter inspired by AeroGen [34] to eliminate low-quality data. This filter evaluates the generated data from two perspectives: 1) semantic consistency assessed by a CLIP model, and 2) layout accuracy verified by a pre-trained ResNet classifier (ensuring alignment between generated objects in bounding boxes and their actual label categories). Finally, we pre-train a detector model on the small-scale training dataset and employ it for active sampling on the filtered data to automatically select the final samples with high value. Notably, this process incorporates a specially designed active sampling mechanism (named AAS) by introducing Attribute-correlated Training Difficulty Factors (ATDF) to optimize downstream detector accuracy.

**Attribute-correlated Training Difficulty Factors.** The proposed ATDF can prompt the detector to select more challenging samples, thereby balancing training difficulty. Specifically, the ATDF is computed during the detector's pretraining phase. We measure each box's accuracy against ground truth, aggregate attribute-specific difficulties, and perform intra-dimensional normalization. In this process, the accuracy of each predicted bounding box is calculated using the method proposed in [4], which can be defined as

$$\text{Acc}(b, \hat{b}) = \hat{p}^{\gamma} \cdot \text{IoU}(b, \hat{b})^{1-\gamma}. \tag{6}$$

Here, $\gamma$ donates a hyper-parameter, $\hat{b}$ and $b$ are the predicted box and the corresponding ground truth, $\hat{p} \in [0, 1]$ is the prediction confidence of $\hat{b}$, the notation $\text{IoU}(b, \hat{b}) \in [0, 1]$ represents the Intersection-over-Union (IoU) metric calculation between $b$ and $\hat{b}$.

Based on the defined bounding box prediction accuracy, we compute the ATDF across all attributes in four dimensions[3]. Each predicted box inherits the extra three specified attributes from its image's category label. Ultimately, each predicted box's accuracy is assigned to the specified attributes across all four dimensions. For the $j$-th evaluation iteration, let $N_s^j$ be the number of predicted boxes possessing the $s$-th attribute. Its initial training difficulty $d_s^j$ can be expressed as

$$d_s^j = \frac{1}{N_s^j} \sum_{n=1}^{N_s^j} (1 - \text{Acc}_n). \tag{7}$$

Then, we employ Exponential Moving Average (EMA) to update the ATDF for each attribute, enabling it to characterize the difficulty discrepancy across the entire training and validation sets. Specifically, the final ATDF of the $s$-th attribute in the $j$-th iteration is obtained by weighting the initial ATDF with the previous timestep's ATDF, expressed as

$$d_s^j \leftarrow m_s^{j-1} d_s^{j-1} + (1 - m_s^{j-1}) d_s^j, \tag{8}$$

with $m_s^{j-1}$ being the attribute-wise momentum at the previous moment. The momentum term for the current iteration $m_s^j$ is updated based on object presence/absence, formalized as

$$m_s^j = \begin{cases} m_s^{j-1}, & \text{If } N_s^j > 0, \\ m_s^0 \cdot m_s^{j-1}, & \text{If } N_s^j = 0, \end{cases} \tag{9}$$

where $m_s^0$ being the initial momentum. This adaptive momentum mechanism accelerates update rates for rare samples, thereby mitigating update speed disparities caused by sample imbalance within the same dimension [42]. Finally, the ATDFs of all attributes within the same dimension are transformed into a probability distribution via the softmax function, which can represent the training difficulty of each attribute in the entire pretraining phase. Note that higher probability values indicate higher training difficulty.

**Attribute-dependent Active Sampling.** We design the AAS to actively select high-value samples from the data pool generated by X-to-Maritime to optimize downstream detection tasks. Specifically, taking the original train set as $X_{\text{train}}$, the filtered generative dataset pool is defined as $X_{\text{gen}}$. The core objective of AAS is to select high-value samples $X_{\text{sel}} \subset X_{\text{gen}}$ using a detector $\mathcal{D}$ pre-trained on $X_{\text{train}}$.

---

[3]Besides the category, we add three additional dimensions, i.e., viewpoint, location, and imaging environment.

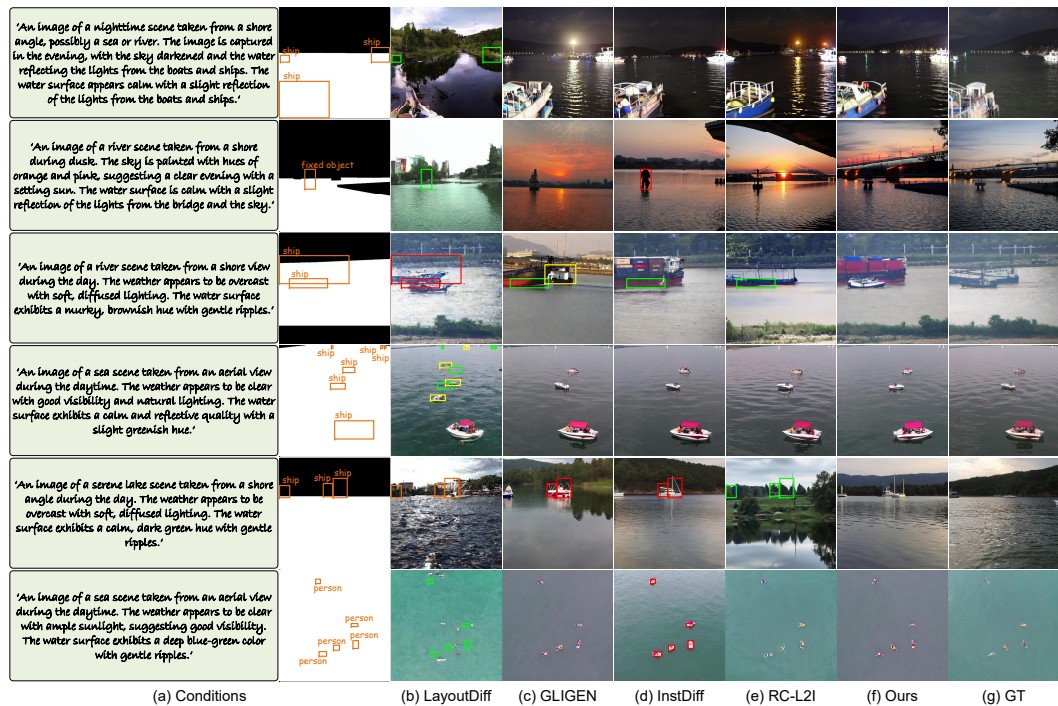

(a) Conditions   (b) LayoutDiff   (c) GLIGEN   (d) InstDiff   (e) RC-L2I   (f) Ours   (g) GT

Figure 4: Comparison of image generation on MGD. The red, green, and yellow bounding boxes indicate low-quality/incorrect generation, missed generation, and unexpected generation, respectively.

Subsequently, both $X_{\text{train}}$ and $X_{\text{sel}}$ are utilized to fine-tune $\mathcal{D}$ further. Specifically, each predicted box is first used to calculate accuracy with its corresponding ground truth label according to Eq. 6. The ATDF values serve as weights for the detection accuracy scores, producing a composite training difficulty measure per image. For a given image with viewpoint, location, and imaging environment ATDFs defined as $d_{\text{view}}$, $d_{\text{loc}}$, and $d_{\text{env}}$, respectively, and with class-wise ATDFs $\{d_{\text{cls}}^n\}_{n=1}^N$ of $N$ objects, the image's training difficulty $d$ is computed as

$$d = \delta \prod_{\alpha \in A} d_\alpha \cdot \frac{1}{N} \sum_{n=1}^N d_{\text{cls}}^n \cdot (1 - \text{Acc}_n), \qquad (10)$$

where $A = \{\text{view}, \text{loc}, \text{env}\}$ and $\delta$ is a tunable parameter. Finally, all samples are ranked by $d$, and the top-k highest-difficulty instances are selected to form the $X_{\text{sel}}$.

## 4 Experiments

### 4.1 Experimental Settings

**Implementation Details.** The Neptune-X framework is implemented in PyTorch 1.13 (Python 3.8) and executed on a PC with 2 Intel(R) Xeon(R) Silver 4410Y CPUs and 4 NVIDIA 5880 Ada GPUs. In the training, we employ the AdamW optimizer with an initial learning rate of $5 \times 10^{-5}$ for 100,000 iterations (requiring $\sim 100$ training hours), while we apply the standard data augmentation techniques, including random horizontal flipping and scale resizing. The patch size and batch size are $512 \times 512$ and 8 for model training. Notably, we reduce training costs and preserve the base model's generative capabilities by freezing the SD weights while only updating layout condition embedders and BiOW-Attn modules.

Table 1: Data source of MGD.

| Source | Imaging Viewpoint | Num. |
|---|---|---|
| MaSTr1325 [3] | ship view | 800 |
| USVInland [6] | ship view | 1000 |
| MIT Sea Grant [9] | ship view | 100 |
| SMD [24] | shore and ship view | 400 |
| Seaships [33] | shore view | 1500 |
| Seagull [29] | aerial view | 2996 |
| Fvessel [12] | shore view | 1500 |
| LaRS [52] | shore, ship, and aerial view | 1973 |
| Others | shore, ship, and aerial view | 1631 |
| MGD | shore, ship, and aerial view | 11900 |

Table 2: FID, CAS, and YOLO Score comparisons of different methods on image generation. The best and second-best results are highlighted in **bold** and underlined.

| Methods | Conditions | Venue & Year | FID ↓ | CAS ↑ | YOLO Score ↑ mAP/mAP$_{50}$/mAP$_{75}$ |
|---------|-----------|--------------|-------|-------|------------------------|
| SD1.5 [30] | Text | CVPR2022 | 27.65 | – | – |
| LayoutDiff [50] | Box | CVPR2023 | 18.17 | 63.77 | 0.83/2.68/0.29 |
| GLIGEN [18] | Text + Box | CVPR2023 | 20.02 | 77.06 | 12.74/30.36/8.99 |
| InstDiff [38] | Text + Box + Mask | CVPR2024 | 19.43 | 76.65 | 12.46/29.73/9.07 |
| RC-L2I [5] | Text + Box + Mask | NeurIPS2024 | 25.63 | 74.84 | 8.75/22.99/5.48 |
| Ours | Text + Box + Mask | NeurIPS2025 | **18.05** | **79.34** | **17.08/39.14/13.52** |

Table 3: mAP and mAP$_{50}$ comparison with/without generated data.

| Model | mAP ↑ | mAP$_{50}$ ↑ |
|-------|-------|--------------|
| YOLOv10 [37] | 39.99 | 61.13 |
| *+Gen Data* | **43.62 (+9.08%)** | **65.50 (+7.15%)** |
| YOLOv11 [16] | 41.29 | 62.51 |
| *+Gen Data* | **44.43 (+7.60%)** | **66.15 (+5.82%)** |
| YOLOv12 [35] | 39.06 | 60.53 |
| *+Gen Data* | **42.91 (+9.86%)** | **63.85 (+5.48%)** |

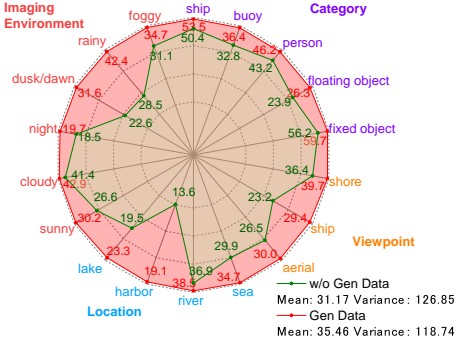

Figure 5: YOLOv11 accuracy improvement visualization across various attributes.

**Ship Generation Dataset.** To advance research on image generation and object detection in maritime scenarios, we propose MGD, a comprehensive maritime image generation dataset. As shown in Table 1, MGD consists of 11,900 samples collected from multiple benchmark datasets and images we captured using various imaging devices, including coastal surveillance systems, UAVs, smartphones, and DSLR cameras. Each sample contains image with corresponding caption, water surface mask, and bounding box annotation, covering five object categories (ship, buoy, person, floating object, and fixed object), three viewpoints (shore-based, shipboard, and aerial), four locations (sea, river, harbor, and lake), and six imaging environments (sunny, cloudy, foggy, rainy, dawn/dusk, and night). Furthermore, MGD is split into training (7,140 samples), validation (2,380 samples), and test sets (2,380 samples) in a 3:1:1 ratio, with the validation and test sets combined for image generation evaluation. More details about MGD are provided in the supplementary materials.

**Evaluation Metrics.** For the image generation evaluation, we use the Frechet Inception Distance Score (FID) [13] for evaluating image generation quality, Classification Score (CAS) [28], and YOLO Score [19] for assessing generated object accuracy. For the data augmentation experiment, the mean Average Precision (mAP) and mAP$_{50}$ are utilized.

## 4.2 Image Generation Experiments

As shown in Table 2, our method demonstrates superior performance across all three evaluation metrics. While LayoutDiff achieves competitive FID scores, its significantly inferior YOLO Score demonstrates limited practical applicability. Notably, our method achieves significant improvements in both CAS (+2.28) and YOLO Score (+4.34/8.78/4.45), the two key metrics evaluating controlled generation capability, substantially outperforming current SOTA approaches. Furthermore, Fig. 4 presents several generated instances of controllable diffusion methods for maritime image generation. These SOTA competitors frequently exhibit missing, erroneous, and unrealistic generation. In contrast, our method achieves superior controllable object generation through the BiOW-Attn module, which enhances object-water interactions to produce more harmonious and realistic maritime scenes.

## 4.3 Data Augmentation Experiments

**Effectiveness on Traditional Detectors.** In this section, we validate the effectiveness of the proposed Neptune-X. Specifically, three advanced object detectors (i.e., YOLOv10 [37], 11 [16], and 12

[35]) were selected for evaluation. Quantitative results on the test set are presented in Table 3. Notably, all detectors demonstrate significant performance improvements by adding generated data as training samples, achieving mAP gains of 7-10% and $mAP_{50}$ improvements of 5-8%. Meanwhile, Fig. 5 demonstrates the detection accuracy improvement of YOLOv11 across all attributes in four dimensions before and after data augmentation. All metrics show improvement, with particularly significant gains observed for attribute categories that originally had lower detection accuracy (The mean increased by 13.77% and the variance decreased by 6.39%)[4]. These results demonstrate the superiority of our proposed generation-selection paradigm for marine object detection, achieving significant accuracy improvement while mitigating cross-attribute training difficulty disparities.

**Effectiveness on Open-Vocabulary Detectors.** The proposed data augmentation method demonstrates broad applicability and can be integrated with various types of detectors, including open-vocabulary detectors. To validate its generalization capability, additional experiments were conducted using Grounding DINO [22]. The experimental results, as shown in Table 4,

Table 4: mAP and $mAP_{50}$ comparison with/without generated data. $^{\dagger}$ denotes fine-tuned on our dataset.

| Model | mAP ↑ | $mAP_{50}$ ↑ |
|---|---|---|
| Grounding DINO | 8.42 | 12.60 |
| Grounding DINO$^{\dagger}$ | 65.03 | 86.12 |
| *+Gen Data* | **68.04 (+4.63%)** | **89.86 (+4.34%)** |

indicate that the proposed method significantly enhances the detection performance, with notable improvements observed in both mAP and mAP50 metrics. These findings suggest that the method is not only compatible with common YOLO-series detectors but also effectively improves the performance of open-vocabulary detectors.

Table 5: Ablation study of different generation configurations.

| ObjCA | WatCA | BiCA | | FID ↓ | CAS ↑ | YOLO Score ↑ |
| | | Obj2WatCA | Wat2ObjCA | | | $mAP/mAP_{50}/mAP_{75}$ |
|---|---|---|---|---|---|---|
| ✓ | | | | 21.44 | 76.23 | 10.69/26.01/6.99 |
| ✓ | ✓ | | | 19.57 | 78.15 | 13.37/29.60/10.78 |
| ✓ | ✓ | ✓ | | 18.35 | 78.00 | 12.52/27.58/10.06 |
| ✓ | ✓ | | ✓ | 18.37 | 78.68 | 15.60/36.13/12.09 |
| ✓ | ✓ | ✓ | ✓ | **18.05** | **79.34** | **17.08/39.14/13.52** |

## 4.4 Ablation Study

**Effectiveness of Generation Modules.** In this subsection, we systematically evaluate different cross-attention (CA) components for maritime scene generation, including Object CA (ObjCA), Water CA (WatCA), and the bidirectional CA (BiCA) that consists of Water-to-Object CA (Wat2ObjCA) and Object-to-Water CA (Obj2WatCA). As quantitatively demonstrated in Table 5, our experiments reveal several key findings. The basic ObjCA alone yields unsatisfactory performance in both overall image quality (evaluated by FID) and target generation accuracy (assessed by

Table 6: Ablation study of different sampling strategies.

| Methods | Number | mAP ↑ | $mAP_{50}$ ↑ |
|---|---|---|---|
| N/O | 0 | 39.99 | 61.13 |
| Random | 5,000 | 41.48 | 63.19 |
| | 10,000 | 43.31 | 64.95 |
| AAS | 5,000 | 43.11 | 64.70 |
| | 10,000 | **43.62** | **65.50** |

the other two metrics) for maritime scenario generation. While introducing water conditions significantly enhances generation capability (notably reducing FID by 1.87), the simple usage fails to properly model object-water interactions, resulting in unrealistic scene-target relationships that limit YOLO Score improvement despite decent CAS results. The bidirectional attention mechanism provides a comprehensive solution. Specifically, Wat2ObjCA effectively improves CAS and YOLO Score by enhancing object features based on water context, while Obj2WatCA further refines water characteristics using object conditions. The full integration of all modules achieves state-of-the-art performance, highlighting the importance of our innovative dual-path modeling that simultaneously ensures high-quality scene generation and physically plausible object-water interactions.

---

[4]It is worth noting a counterintuitive observation: detection performance under sunny conditions was lower, likely due to intense sunlight causing water surface glare, which is uncommon in typical training data.

**Effectiveness of AAS.** To evaluate the effectiveness of the proposed Attribute-correlated Active Sampling (AAS) strategy, we conducted comparative experiments using YOLOv10 as the baseline detector. As shown in Table 6, the results demonstrate AAS's clear advantages over random sampling through two key observations. First, AAS achieves superior performance. The mAP and $mAP_{50}$ obtained using 5,000 AAS-selected samples match that of 10,000 randomly sampled instances, while significantly higher than the version of 5,000 randomly sampled instances. Second, while random sampling shows substantial performance gains when increasing from 5,000 to 10,000 samples ($\Delta mAP=1.83$ and $\Delta mAP_{50}=1.76$), AAS

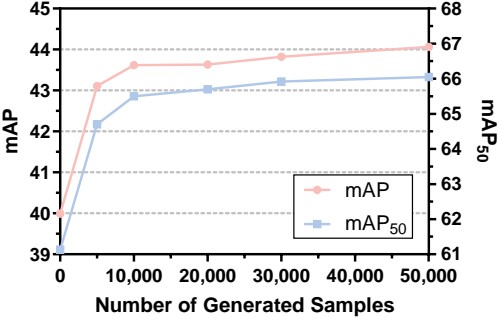

Figure 6: Correlation between detection accuracy and the number of generated samples used.

exhibits minimal improvement ($\Delta mAP=0.51$ and $\Delta mAP_{50}=0.80$) in this range. This difference stems from AAS's targeted selection mechanism, which effectively identifies and prioritizes high-value samples in the sampling phase, leading to faster convergence and reduced need for additional samples. To thoroughly validate this and determine the appropriate number of training samples, this section conducted relevant experiments. As shown in Fig. 6, a significant saturation effect in detection performance improvement was observed when the data volume increased from 10,000 to 20,000 samples. This phenomenon indicates that the AAS method achieves performance gains by pre-screening the most valuable generated samples. For lower-ranked samples, since the model has already learned relevant features from previous high-value samples, the information contained in these samples is no longer novel to the model. Therefore, further increasing such samples does not lead to significant performance improvements. The active selection strategy of the AAS method enables rapid performance gains with a smaller data volume while significantly reducing computational costs and additional training overhead. This stands in sharp contrast to traditional methods that rely on large amounts of data. To sum up, these results collectively confirm the efficiency and practicality of the AAS method in maritime scene object detection tasks.

# 5 Conclusion, Limitation, and Future Work

In this paper, we have presented a data generation-selection paradigm (Neptune-X) to reduce the cost of data collection and annotation while addressing cross-attribute training difficulty caused by limited sample diversity. Our method intends to enhance maritime scene generation through attention to object–water interaction and improve training efficiency via an attribute-aware sample selection strategy that considers both predicted accuracy and difficulty priors. We have also constructed a new dataset to support maritime image generation. Extensive experiments have demonstrated the effectiveness of our approach in both image synthesis and data augmentation, significantly boosting the detection performance.

While our method demonstrates significantly better performance, it currently relies on a fixed set of predefined attribute categories (e.g., viewpoint, lighting condition, object type) to estimate training difficulty. This discrete formulation may limit the granularity of difficulty modeling and adaptation. Future work should explore the framework extension to support continuous or hierarchical attribute spaces, allowing for more nuanced difficulty estimation.

**Acknowledgments.** This work is supported by the JC STEM Lab of Smart City funded by The Hong Kong Jockey Club Charities Trust (2023-0108), the Hong Kong SAR Government under the Global STEM Professorship and Research Talent Hub, the Guangdong Natural Science Funds for Distinguished Young Scholars (Grant 2023B1515020097), the National Research Foundation Singapore under the AI Singapore Programme (AISG Award No: AISG4-TC-2025-018-SGKR), and the Lee Kong Chian Fellowships.

**Impact Statement.** This research advances machine learning and develops innovative solutions supporting technological progress in areas like intelligent navigation and smart infrastructure. It is not oriented toward commercial exploitation or other non-academic applications.

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

# A  Appendix / Supplemental Material

## A.1  Overview

This supplement provides more data and method details as well as experimental results, including:

- We provide detailed information about the Maritime Generated Dataset (MGD).
- We offer more details about the proposed Neptune-X.
- We conduct more experiments to verify the effectiveness and superiority of the proposed method.

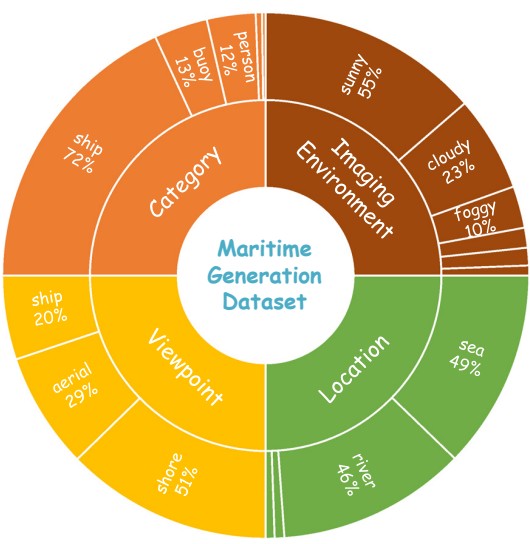

Figure 7: The percentages of various dimensions and attributes in our MGD dataset.

Table 7: Sample numbers and percentages of various dimensions and attributes.

| Dimensions | Attributes | Number | Proportion |
| --- | --- | --- | --- |
| Category | ship | 29313 | 72.44% |
| | buoy | 5326 | 13.16% |
| | person | 4843 | 11.97% |
| | floating obj. | 618 | 1.53% |
| | fixed obj. | 366 | 0.90% |
| View | shore | 6042 | 50.77% |
| | ship | 2459 | 20.66% |
| | aerial | 3399 | 28.56% |
| Location | sea | 5829 | 48.98% |
| | river | 5531 | 46.48% |
| | harbor | 282 | 2.37% |
| | lake | 258 | 2.17% |
| Imaging Environment | sunny | 6491 | 54.55% |
| | cloudy | 2794 | 23.48% |
| | foggy | 1225 | 10.29% |
| | rainy | 515 | 4.33% |
| | dawn/dusk | 583 | 4.90% |
| | night | 292 | 2.45% |

## A.2  Maritime Generation Dataset

We constructed the Maritime Generation Dataset (MGD), the first generation dataset for maritime scenarios. In particular, the MGD contains 11,900 samples covering diverse semantic scenes. Fig. 7 and Table 7 illustrate the numbers and percentages of samples of MGD in terms of viewpoint, location, imaging environment, and object category. Furthermore, each image sample contains visual images with corresponding labels, including water surface mask, object bounding boxes, and multi-level descriptions of the entire image, water surface, and objects. More specifically, the flowchart of data labelling is shown in Fig. 8, which contains two steps: data collection and data annotation.

**Data Collection.** The data collection process involves the utilization of imaging devices deployed across multiple platforms. During this phase, we strategically selected and leveraged multiple open-source maritime benchmarks while employing diverse imaging equipment, including surveillance cameras, DSLR cameras, and smartphone cameras, to gather large-scale raw data. The collected datasets exhibit significant variations in geographical capture locations, temporal acquisition parameters, meteorological conditions, water surface characteristics, and surface target typologies. This systematic diversity guarantees the comprehensive data richness, thereby enabling generative models to synthesize highly diversified maritime scenarios.

**Data Annotation.** To optimize annotation efficiency, we employed state-of-the-art image detectors, segmenters, and vision-language models for assisted annotation, with human annotators performing calibration and verification of the generated labels. In the first stage, the T-Rex2 [15] and SAM2 [27] models are utilized to generate target bounding boxes and water surface masks, respectively. This semi-automated process requires manual input: exemplar images for T-Rex initialization and point prompts for SAM2 segmentation. All model outputs undergo secondary human verification to ensure annotation accuracy. The second stage leverages the state-of-the-art LLaVA-Next [17]

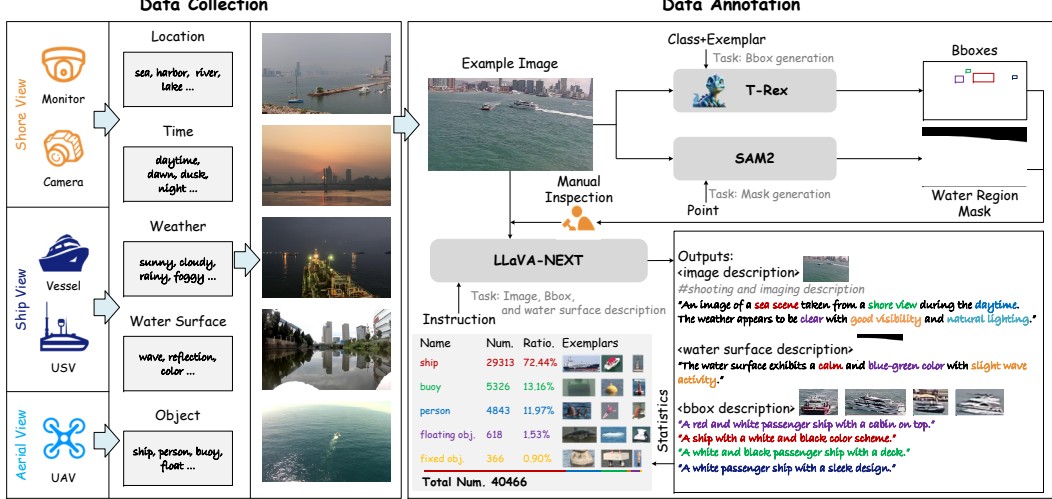

Figure 8: Flowchart of data labelling.

---

**Algorithm 1** BiOW-Attn

---

**Input:** Input features $f_{in}$, object's embeddings $\{\mathcal{C}_o^i\}_{i=1}^O$, objects' masks $\{\mathcal{M}_o^i\}_{i=1}^O$, water surface embedding $\mathcal{C}_w$, water surface mask $\mathcal{M}_w$, learnable null object/water surface embeddings $\{\mathbf{null}_{obj}, \mathbf{null}_{wat}\}$, learnable object/water surface gated scaler $\{\beta_o, \beta_w\}$.

1: $f_{in} = \text{reshape}(\mathbf{F}_{in})$.
2: Take $\mathcal{C}_o^i$ and $f_{in}$ as inputs and calculate the object-guided feature $f_o^i$ via Eq. (4).
3: Take $\{f_o^i\}_{i=1}^O$, $\{\mathcal{M}_o^i\}_{i=1}^O$, and $\mathbf{null}_{obj}$ as inputs and get final object-guided feature $\mathbf{F}_o$ via Eq. (5).
4: Take $\mathcal{C}_w$ and $f_{in}$ as inputs and calculate the water surface-guided feature $f_w$ via Eq. (4).
5: Take $f_w$, $\mathcal{M}_w$, and $\mathbf{null}_{wat}$ as inputs and get final water surface-guided feature $\mathbf{F}_w$ via Eq. (5).
6: Perform bidirectional attention to obtain $\mathbf{F}_o'$ and $\mathbf{F}_w'$.
7: $f_{out} = f_{in} + \tanh(\beta_o) \cdot \mathbf{F}_o' + \tanh(\beta_w) \cdot \mathbf{F}_w'$.
8: $\mathbf{F}_{out} = \text{reshape}(\text{MLP}(f_{out}))$.
9: **return** $\mathbf{F}_{out}$.

---

vision-language model for multi-scale scene understanding. Through tailored text prompts, the model analyzes three distinct levels of visual features:

- **Image Description** includes shooting scenarios, camera perspectives, timestamps, weather conditions, and lighting.

- **Water Surface Description** contains surface calmness, color properties, and wave patterns.

- **Object Description** documents color attributes and detailed category features (e.g., specific ship classifications)[5].

### A.3 More Details of Neptune-X

**Bidirectional Object-Water Attention.** The detailed process of the proposed BiOW-Attn module is shown in Algorithm 1. Specifically, we first reshape the input features $\mathbf{F}_{in}$ to facilitate cross-attention computation (line 1). We then perform spatially masked cross-attention operations via Eq. (4) and (5) to obtain object-conditioned feature $\mathbf{F}_o$ and water-conditioned feature $\mathbf{F}_w$ respectively (lines 2-5). A bidirectional cross-attention module subsequently models interaction relationships between the water surface object and the aquatic surrounding, followed by gated residual fusion (lines 6-7). During initial training, we set $\beta_o = \beta_w = 0$ to ensure fine-tuning stability. Finally, a Feed-Forward Network (FFN) processes the fused features, with output obtained through final reshaping (lines 8-9).

---

[5]Note that these fine-grained object features are extracted for more controllable image generation. Thus, this type of label is excluded from the data generation pipeline to promote randomness in the generated objects.

---

**Algorithm 2** Data Sampling

---

**Input:** X-to-Maritime generator $G$, text condition $\mathcal{C}$, object conditions $\{\mathcal{C}_o^i, \mathcal{M}_o^i\}_{i=1}^O$, water surface condition $\{\mathcal{C}_w, \mathcal{M}_w\}$, number of samples $N$, selected sample collection $X_{\mathbf{sel}}$, ResNet classifier $\zeta$, CLIP text/visual encoder $\{\xi, \xi_v\}$, pre-trained detetcor $\mathcal{D}$.

1: **for** $n = 1, ..., N$ **do**
2:      $I_{\mathbf{gen}} = G(\mathcal{C}, \{\mathcal{C}_o^i, \mathcal{M}_o^i\}_{i=1}^O, \{\mathcal{C}_w, \mathcal{M}_w\})$.
3:      Calculate layout accuracy $\text{Acc}_l$ between $\zeta(I_{\mathbf{gen}})$ and category labels $cls$ corresponding to $\{\mathcal{C}_o^i\}_{i=1}^O$.
4:      Calculate semantic accuracy $\text{Acc}_s$ between $\xi_v(I_{\mathbf{gen}})$ and $\xi(cls)$.
5:      **if** $\text{Acc}_l > \tau_l$ and $\text{Acc}_s > \tau_s$ **then**
6:          Get the predicted bounding boxes via $\mathcal{D}(I_{\mathbf{gen}})$.
7:          Calculate the training difficulty $d_n$ of $n$-th samples via Eq. (10).
8:      **end if**
9: **end for**
10: Sort by $d$ and get the sorted sample set.
11: Select the top-$k$ samples and put them into $X_{\mathbf{sel}}$.
12: **return** $X_{\mathbf{sel}}$

---

**High-quality Data Generation.** During the data generation phase, we enrich the generated samples by combining layout conditions and text caption conditions, ultimately producing 100,000 generated images. Subsequently, we perform data filtering through two key evaluation metrics: layout similarity and semantic similarity.

For layout similarity assessment, we train a ResNet-based classifier on the MGD dataset to evaluate the generated samples. Meanwhile, we employ the CLIP model to compute the cosine similarity between each image and its corresponding textual object descriptions as the semantic similarity metric. For example, if the image contains a ship and a person, then the description is 'an image of a ship and a person'.

In the active sampling stage, we first use the pre-trained detector to identify objects in the images and calculate accuracy by comparing them with the bounding boxes specified in the layout conditions. We then introduce the attribute-related training difficulty factors (ATDFs) as weighting coefficients. Finally, we rank the results and filter underperforming samples into a training pool for iterative model optimization. The detailed flowchart is illustrated in Algorithm 2.

### A.4 Experiments Results

**Evaluation Metrics.** This section introduces the mAP metric used in object detection and three other indicators (FID, CAS, and YOLO Score) used in image generation.

- **mAP and mAP$_{50}$:** Mean Average Precision (mAP) serves as the core evaluation metric for object detection tasks. The calculation process involves four key steps. First, True Positives (TP) and False Positives (FP) are determined based on the Intersection-over-Union (IoU) threshold. Then, detection results are sorted by confidence scores to plot the Precision-Recall (P-R) curve. The area under the P-R curve is computed to obtain Average Precision (AP) for each class. Finally, the mean of AP values across all classes yields the mAP. The PASCAL VOC benchmark employs a fixed IoU threshold of 0.5 (denoted as mAP$_{50}$), while the COCO dataset adopts averaged results across IoU thresholds ranging from 0.5 to 0.95 (denoted as mAP). Compared to mAP$_{50}$, mAP imposes more stringent localization accuracy requirements. These two metrics comprehensively reflect a model's detection stability across categories and its localization precision.

- **Fréchet Inception Distance (FID):** FID [13] is a basic metric for generative model evaluation, measuring the statistical distribution discrepancy between generated images and ground truth in the latent space. In particular, this process can be divided into two steps, i.e, feature extraction and distribution distance calculation. FID first extracts the latent feature vectors $f_{\text{gen}}$, $f_{\text{gt}}$ of generated and real images by Inception-v3, and calculates the mean $\{\mu_{\text{gen}}, \mu_{\text{gt}}\}$ and covariance matrix $\{\Sigma_{\text{gen}}, \Sigma_{\text{gt}}\}$. Finally, the FID distance can be calculated by

$$\text{FID} = \|\mu_{\text{gen}} - \mu_{\text{gt}}\|^2 + \text{Tr}\left(\Sigma_{\text{gen}} + \Sigma_{\text{gt}} - 2(\Sigma_{\text{gen}}\Sigma_{\text{gt}})^{1/2}\right), \tag{11}$$

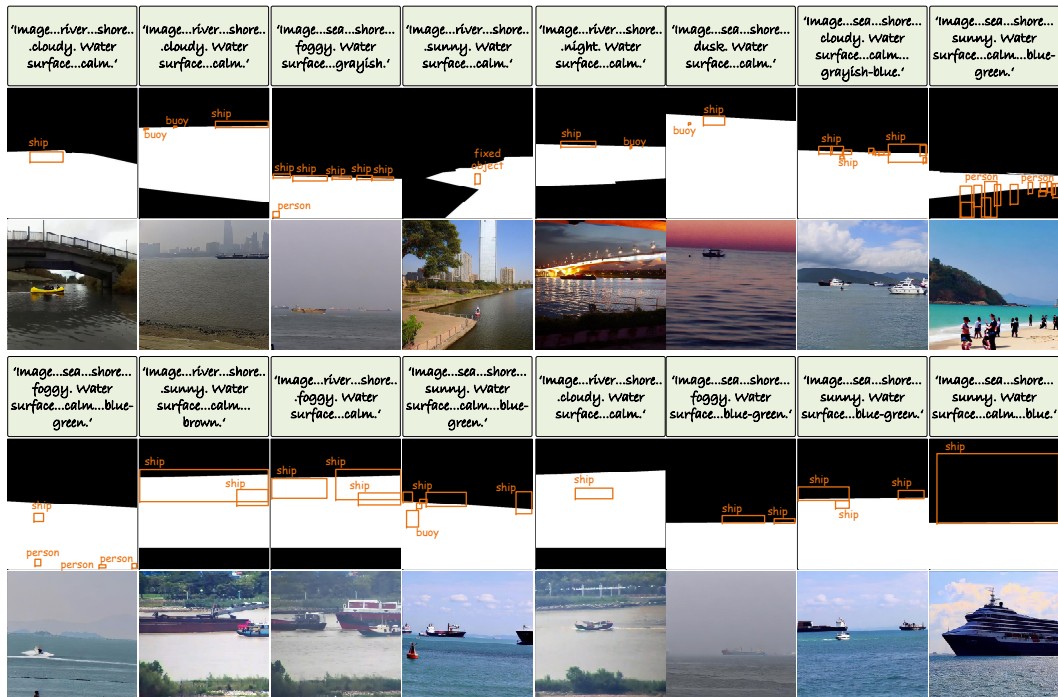

Figure 9: Image generation cases on shore viewpoints.

where $\| \cdot \|^2$ denotes the squared Euclidean norm, $\text{Tr}(\cdot)$ is the matrix trace operator.

- **Classification Score (CAS):** CAS [28] serves as a critical metric for evaluating the generation quality within bounding box-based object constraints. To compute CAS, we first train a ResNet-101 classifier on our MGD for 100 epochs. The trained model is then used to evaluate classification accuracy by comparing the categories predicted by ResNet-101 on generated images against the ground-truth categories specified in the input object conditions.

- **YOLO Score:** YOLO Score [19] is used to evaluate the location and category accuracy of the generated objects. In particular, we utilize a YOLOv10 model trained on our MGD for 100 epochs. This trained detector evaluates generated samples by computing three standard object detection metrics, i.e., mAP (averaged over IoU thresholds from 0.5 to 0.95), $\text{mAP}_{50}$ (using 0.5 IoU threshold), and $\text{mAP}_{75}$ (using 0.75 IoU threshold), which collectively form the YOLO Score.

**More Generation Results.** To fully demonstrate the powerful image generation capabilities of our proposed X-to-Maritime framework, we present more visualization results. As shown in Figs. 9 and 10, we display generated maritime scenes from shore, ship, and aerial perspectives, respectively. Notably, our model demonstrates accurate comprehension of input multi-modality conditions. The generation process is jointly guided by both textual caption conditions (for scene description) and layout conditions (for controlling object and water surface position and content). Most significantly, the framework faithfully reproduces hydrodynamic interactions between water surface objects and their aquatic surroundings while maintaining strict physical plausibility. This capability directly stems from our novel bidirectional object-water cross-attention mechanism, which effectively models the mutual influences between maritime entities and their environment.

In addition, Fig. 11 presents two cases with different random seeds and the removal of text conditions only. The results demonstrate that under identical input settings, both the generated objects and the overall background exhibit rich diversity. This further validates the diversity of the generated results produced by the proposed model, an advantage primarily attributed to the rich semantic features encompassed in the constructed MGD maritime generation dataset.

**More Results of Different Generation Configurations.** As shown in Fig. 12, we compare the generation quality of models under different configurations. It can be clearly observed that using only the ObjCA module, while providing effective object control, fails to account for aquatic environments, leading to unsatisfactory generation results. Representative examples include unrealistic water-object

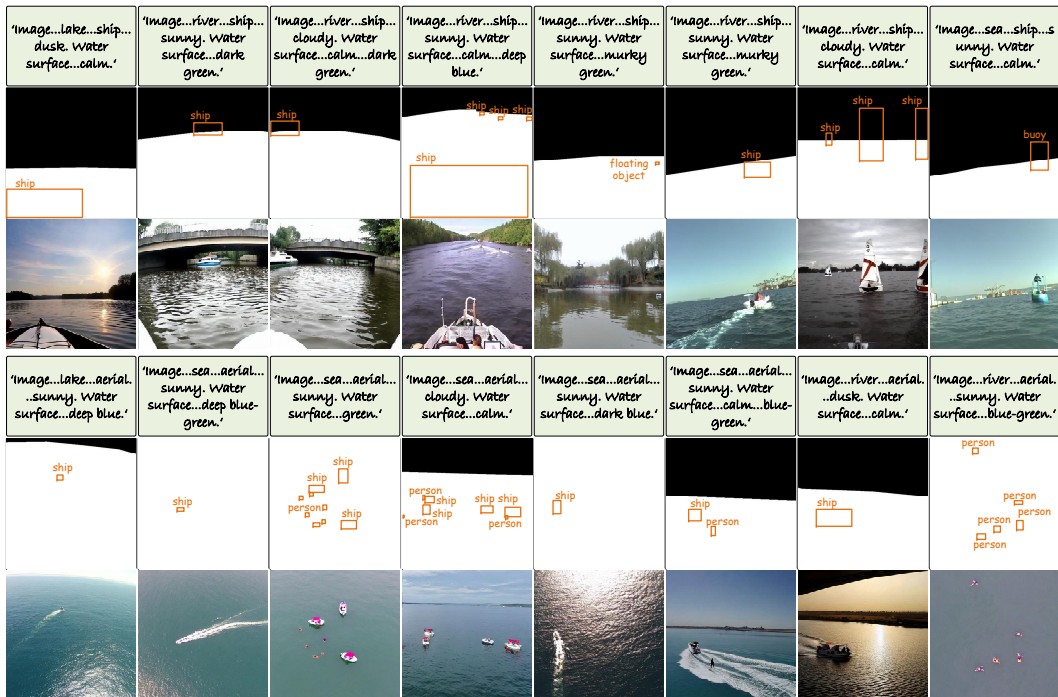

Figure 10: Image generation cases on ship and aerial viewpoints.

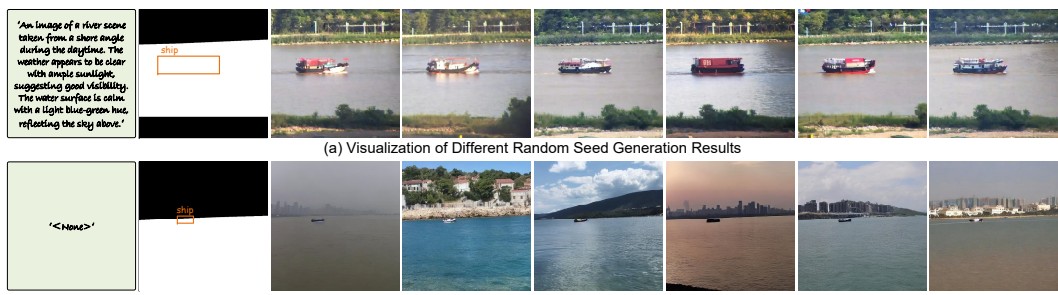

(a) Visualization of Different Random Seed Generation Results

(b) Visualization of Text-Free Conditional Generation Results

Figure 11: Image generation cases using (a) different random seeds and (b) only removing text conditions. The main reason for the scene similarity in (a) is that the text specifies background and hydrological conditions, while the unspecified objects exhibit diversity.

interactions in the second case and ships floating mid-air in the third generated sample. In contrast, introducing water surface conditions significantly alleviates these abnormal generation cases. The Obj2WatCA module enhances generation quality by improving water realism through object-to-water influence. However, the object control precision becomes reduced. This trade-off is visible in the first case, where objects with inaccurate positions appear in the generated results. Meanwhile, using only Wat2ObjCA improves visual target generation quality but still produces build failures. Ultimately, by integrating the advantages of all modules, our method demonstrates superior performance in simulating realistic object-water interactions while maintaining high-fidelity generation.

**Comparison of Different Generators in Data Augmentation.** This section aims to validate the effectiveness of the proposed X-to-Maritime framework in maritime object detection by comparing the performance improvements achieved through different generated data. The generated data were then combined with training data to fine-tune object detection models. Specifically, all generative models utilized identical conditional inputs to generate expanded data. These data, along with original data, were used for fine-tuning the detectors to ensure fairness and comparability of experimental results. In the experiments, YOLOv10 [37] was employed as the object detection model, with mAP and mAP50 serving as evaluation metrics.

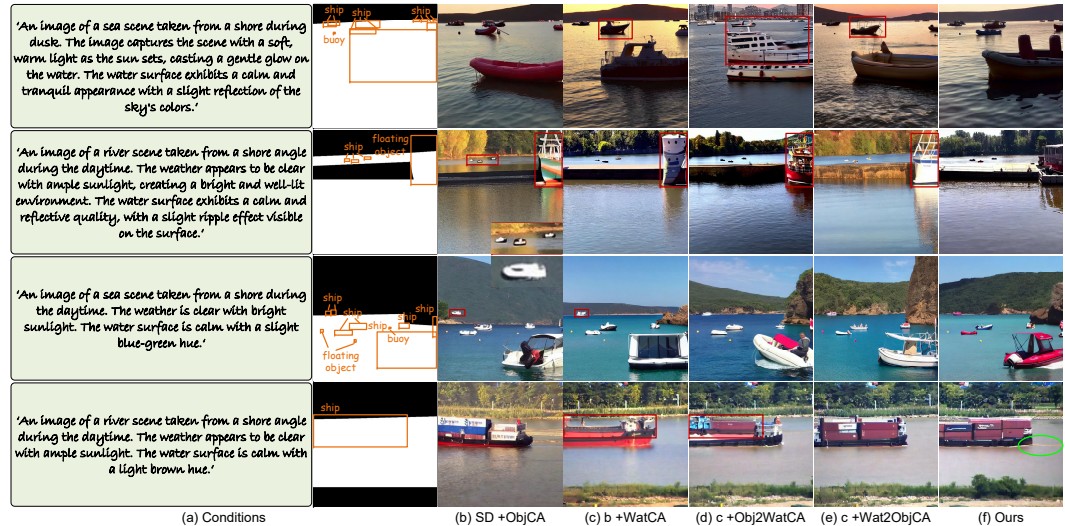

| | 'An image of a sea scene taken from a shore during dusk. The image captures the scene with a soft, warm light as the sun sets, casting a gentle glow on the water. The water surface exhibits a calm and tranquil appearance with a slight reflection of the sky's colors.' | 'An image of a river scene taken from a shore angle during the daytime. The weather appears to be clear with ample sunlight, creating a bright and well-lit environment. The water surface exhibits a calm and reflective quality, with a slight ripple effect visible on the surface.' | 'An image of a sea scene taken from a shore during the daytime. The weather is clear with bright sunlight. The water surface is calm with a slight blue-green hue.' | 'An image of a river scene taken from a shore angle during the daytime. The weather appears to be clear with ample sunlight. The water surface is calm with a light brown hue.' |

|          (a) Conditions          |          (b) SD +ObjCA          |          (c) b +WatCA          |          (d) c +Obj2WatCA          |          (e) c +Wat2ObjCA          |          (f) Ours          |

Figure 12: Comparison of different configurations in image generation.

Table 8: Comparison of different data generation methods on YOLOv10 detection accuracy.

| Methods | Venue & Year | mAP $\uparrow$ | mAP$_{50}$ $\uparrow$ |
|---|---|---|---|
| w/o | | 39.99 | 61.13 |
| LayoutDiff [50] | CVPR2023 | 40.03 | 61.01 |
| GLIGEN [18] | CVPR2023 | 41.54 | 62.85 |
| InstDiff [38] | CVPR2024 | 41.32 | 62.57 |
| RC-L2I [5] | NeurIPS2024 | 41.48 | 63.26 |
| Ours | NeurIPS2025 | **43.62** | **65.50** |

Table 9: Comparison of different sampling methods on YOLOv10 detection accuracy.

| Methods | mAP $\uparrow$ | mAP$_{50}$ $\uparrow$ |
|---|---|---|
| w/o | 39.99 | 61.13 |
| Entropy | 42.62 | 64.40 |
| Variance | 42.42 | 64.17 |
| Margin | 42.87 | 64.55 |
| Greedy K-Center | 42.24 | 63.90 |
| K-Means Corset | 42.27 | 63.79 |
| AAS | **43.62** | **65.50** |

The experimental results, shown in Table 8, indicate that existing generative methods perform poorly in maritime scenarios. Due to their failure to adequately account for the unique characteristics of maritime environments, such as complex interactions between water bodies and objects, the synthetic data generated by these methods exhibit significant shortcomings in realism and detail fidelity. In contrast, the proposed X-to-Maritime framework incorporates a Bidirectional Object-Water Attention (BiOW-Attn) module to model the water-object interaction. This module effectively captures the influence of water on object appearance and the feedback effects of objects on water, thereby generating high-quality results. Experimental results demonstrate that the generated data produced by the X-to-Maritime framework significantly enhance detection accuracy.

**Comparison of Different Sampling Methods in Data Augmentation.** To comprehensively demonstrate the advantages of the proposed AAS method, extensive comparative experiments were conducted with five different active learning approaches. These methods include uncertainty-based sampling strategies (Entropy, Variance, and Margin) and diversity-based sampling strategies (Greedy K-Center and K-Means Corset). For the diversity-based methods, a 7-dimensional feature vector was constructed, incorporating the number of detection boxes, average detection box area, standard deviation of detection box areas, average confidence score, standard deviation of confidence scores, mean x-coordinate of detection boxes, mean y-coordinate of detection boxes, and the number of object categories. The experimental results, presented in Tables 9, show that while traditional active learning methods can improve object detection performance to some extent, the proposed AAS method achieves significantly greater enhancement by comprehensively considering attributes specific to maritime scenarios, such as water conditions, weather states, and viewpoint information. This validates the effectiveness of AAS in selecting the most valuable synthetic samples through the integration of multi-dimensional attribute factors, further demonstrating its superiority in visual perception for maritime intelligent transportation systems.

Table 10: Ablation study of different generation configurations on detection performance.

| ObjCA | WatCA | BiCA Obj2WatCA | Wat2ObjCA | mAP ↑ | mAP$_{50}$ ↑ |
|:---:|:---:|:---:|:---:|:---:|:---:|
| ✓ | | | | 40.94 | 61.18 |
| ✓ | ✓ | | | 41.09 | 61.15 |
| ✓ | ✓ | ✓ | | 42.54 | 63.85 |
| ✓ | ✓ | | ✓ | 42.75 | 63.93 |
| ✓ | ✓ | ✓ | ✓ | **43.62** | **65.50** |

Table 11: Ablation study of ATDFs on detection performance.

| Viewpoint | Location | Imaging Environment | Object Category | mAP ↑ | mAP$_{50}$ ↑ |
|:---:|:---:|:---:|:---:|:---:|:---:|
| ✓ | | | | 40.26 | 62.09 |
| ✓ | ✓ | | | 41.27 | 62.26 |
| ✓ | ✓ | ✓ | | 41.40 | 63.10 |
| ✓ | ✓ | ✓ | ✓ | **43.62** | **65.50** |

**Ablation Study of Data Augmentation with Different Generation Configurations.** This section conducts an ablation study on each component of the BiOW-Attn module to quantitatively analyze their individual contributions to the improvement of object detection performance, as shown in Table 10. The experimental results demonstrate that solely introducing water conditions can enhance object detection performance to some extent, but such improvement remains limited. In contrast, the bidirectional attention module, by simulating the interactions between water bodies and objects, more accurately reflects the physical characteristics of maritime scenarios, thereby achieving greater performance gains in object detection.

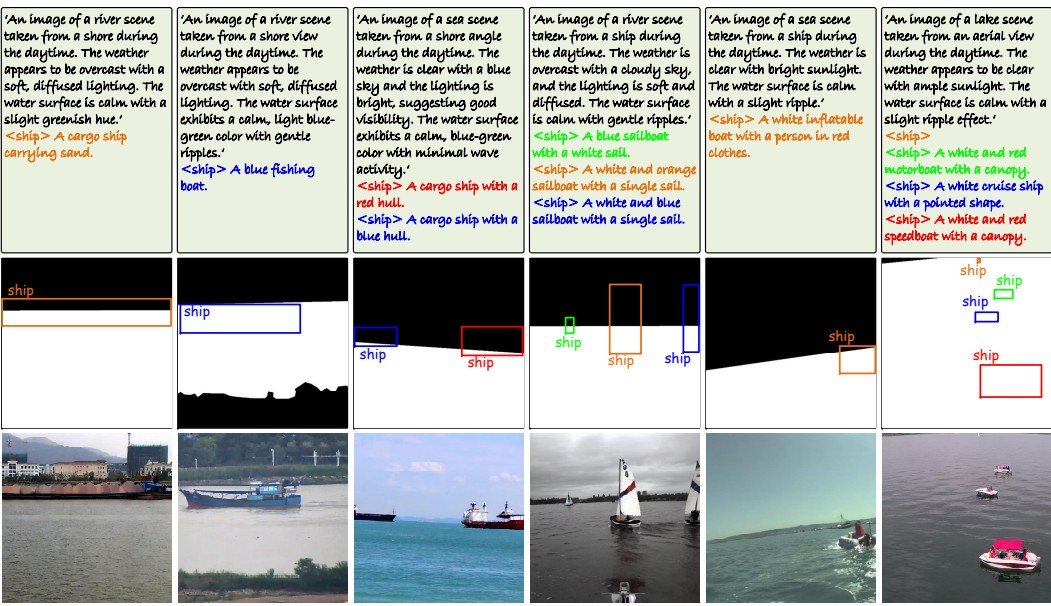

Figure 13: Image generation cases via fine-grained object control.

**Ablation Study of ATDFs.** In maritime object detection tasks, the diversity and quality of samples are crucial for the model's generalization capability. To more effectively leverage generated data, we propose an Attribute-correlated Active Sampling (AAS) strategy based on Attribute-correlated Training Difficulty Factors (ATDFs). This strategy quantifies the importance of each sample across different attribute dimensions and prioritizes the selection of samples that contribute most to improving object detection performance. To further investigate the role of ATDFs in sample selection, this section validates the effectiveness of each dimensional ATDF in object detection tasks. As shown in Table 11, the contributions of different attribute-dimensional factors to object detection performance are presented. Each attribute-dimensional factor is associated with specific characteristics of

maritime scenarios. The experimental results indicate that as more attribute-dimensional factors are incorporated, the sample selection strategy becomes more precise in identifying the most valuable samples for enhancing detection performance. Specifically, when only a single-dimensional factor is used, the improvement in object detection performance is relatively limited. However, as the number of dimensional factors increases, the accuracy of sample selection significantly improves, and object detection performance demonstrates a gradual upward trend. This suggests that combining multi-dimensional attribute factors can more comprehensively capture the complexity and diversity of samples, thereby providing more valuable training data for detection models.

**Discussion on More Controllable X-to-Maritime.** In our data generation task, to enhance the diversity of generated targets, we employed only the object category as the object embedding feature. To explore more fine-grained control over the generated targets, including detailed category specifications (e.g., ship types like sailboats, inflatable boats, cargo ships, fishing ships) and visual attributes like color, we concatenate both the object category and its detailed description to form a more controllable object embedding for model training. As demonstrated in Fig. 13, the trained model exhibits clear awareness of multi-level textual descriptions (at the image level, water surface level, and object level), while accurately simulating realistic maritime scenes according to layout conditions. The generated samples convincingly show the model's capability to respond to hierarchical textual controls while maintaining photorealistic quality.

