# OpenReview forum: "Neptune-X: Active X-to-Maritime Generation for Universal Maritime Object Detection"
_NeurIPS.cc/2025/Conference — NeurIPS 2025 spotlight_

### Official Review · Reviewer_1w7F · 2025-06-25

**Clarity:** 3
**Significance:** 2
**Originality:** 2
**Rating:** 4
**Confidence:** 2

**Summary:**

This paper proposes Neptune-X, a data-centric framework that improves maritime object detection by combining synthetic data generation with task-aware sample selection. It introduces X-to-Maritime, a multi-modality generative model with a Bidirectional Object-Water Attention module for realistic scene synthesis, and Attribute-correlated Active Sampling to enhance training efficiency. The authors also present the Maritime Generation Dataset to support benchmarking. Experiments show significant improvements in both image synthesis and detection performance.

**Questions:**

NA

**Ethical Concerns:**

["NO or VERY MINOR ethics concerns only"]

**Final Justification:**

The rebuttal offers reasonable points about the suitability of this manuscript for NeurIPS. Since I am not an expert in this specific domain, I am no longer inclined to reject the paper and instead support the evaluations of the other reviewers.

**Limitations:**

One potential limitation that is not discussed in the paper is the dual-use nature of the proposed techniques. Although the work is positioned for maritime safety and monitoring, the enhanced object detection and scene synthesis capabilities could also be adapted for military surveillance or defense-related applications. It would be helpful if the authors could briefly acknowledge this potential and clarify the intended scope of deployment.

**Quality:**

3

**Strengths And Weaknesses:**

Strengths

1.	Leveraging image generation for data augmentation is a valuable and promising approach to expand the training set for maritime object detection.

2.	The paper is well-written and easy to follow.

3.	The experiments demonstrate high-quality image synthesis and substantial improvements in detection performance.

Weaknesses

1.	The idea of using image generation for data augmentation is not novel for maritime detection [C] and other domain-specific detection tasks. Prior works such as DriveDreamer [A] and Panacea [B] in autonomous driving have explored similar approaches and evaluated their effectiveness for improving training utility.

2.	The technical contributions are highly domain-specific, and some components—such as the AAS—appear hand-crafted. While these designs are effective within the maritime context, they lack generality and may offer limited methodological insights for the broader machine learning community. As a result, the work may be better suited to a domain-focused venue rather than a general-purpose conference like NeurIPS.

[A] Wang, Xiaofeng, et al. "DriveDreamer: Towards Real-World-Drive World Models for Autonomous Driving." ECCV 2024

[B] Wen, Yuqing, et al. "Panacea: Panoramic and controllable video generation for autonomous driving." CVPR. 2024.

[C] Tang, Datao, et al. "AeroGen: Enhancing remote sensing object detection with diffusion-driven data generation." CVPR. 2025.

---

> ### Author Rebuttal · Authors · 2025-07-29
>
> We sincerely appreciate Reviewer 1w7F's valuable feedback. Our responses to the weaknesses, questions, and limitations are listed below.
>
> **R-W1 Data augmentation explored before for maritime detection** We respectfully disagree with this comment. To our knowledge, maritime scene generation for data augmentation has not been explored in any prior work. While image generation has been applied in other domains such as autonomous driving (the suggested [A, B]) and aerial imagery (the suggested [C]), these approaches are fundamentally incompatible with the maritime setting.
>
> The cited methods assume structured environments with stable, predictable spatial layouts and well-defined object categories. In contrast, maritime scenes are inherently unstructured, with highly variable sea states, ambiguous object boundaries, and complex object-water interactions that lack reliable contextual anchors. These unique challenges render existing approaches ineffective when applied to this domain.
>
> Moreover, our empirical results in Table 2 and Figure 4 clearly demonstrate that general or cross-domain augmentation strategies fail to produce meaningful performance gains in maritime detection. This reinforces the need for a domain-specific generation framework, which our work is the first to provide. Our approach directly addresses the critical aspects of maritime imagery that existing methods overlook, establishing a new and necessary direction for data-driven learning in this field.
>
> **R-W2 Domain-specificity** We respectfully disagree with this comment. The domain-specific nature of our work should be seen as a strength, not a limitation. Addressing complex real-world problems, such as maritime object detection, requires deep domain expertise, problem-specific modeling, and careful attention to context-specific challenges. These qualities reflect the type of thoughtful, impactful work that NeurIPS has a long history of supporting.
>
> It is a misconception to view NeurIPS as a venue limited to general-purpose algorithms. In reality, a significant portion of NeurIPS papers are domain-specific, leveraging machine learning to advance understanding in specialized areas. For example, NeurIPS regularly publishes work on medical images (e.g., tumor segmentation, radiology interpretation), astronomical imaging (e.g., telescope-based object detection), biological microscopy, remote sensing, and robotic perception. These works often include components tailored to their domains, because the structure and semantics of the data vary substantially across application areas.
>
> Our work is in this tradition. Maritime images differ drastically from more commonly studied domains. They are visually ambiguous, lack structured background cues, and involve dynamic interactions between objects and water under varied environmental conditions. Developing effective solutions in this space necessarily requires specialized design choices, and our method was built to address those specific challenges directly.
>
> In this sense, domain-specificity should not be equated with limited relevance. Rather, it reflects a deliberate effort to make machine learning effective in contexts where naive application of general tools would fail. We believe this aligns with the inclusive and applied spirit of NeurIPS.
>
> **R-L Dual-use nature** We appreciate the reviewer's concern about the dual-use potential of our technology. Our proposed model is specifically designed for intelligent navigation and smart maritime applications, similar to autonomous driving and smart city scenarios, and is not intended for military purposes.
>
> Nonetheless, we recognize the importance of addressing responsible deployment. In the final version of the paper, we will include a statement clarifying the intended scope of use and reinforcing our commitment to ethical and non-military applications to help mitigate risks of unintended misuse.

---

> > ### Comment · Reviewer_1w7F · 2025-08-05
> >
> > Thanks for the rebuttal. I will raise my score to 4.

---

### Official Review · Reviewer_tKB5 · 2025-06-28

**Clarity:** 4
**Significance:** 3
**Originality:** 2
**Rating:** 5
**Confidence:** 4

**Summary:**

- The authors propose Neptune-X1, a unified framework for maritime object detection that combines diverse scene generation with task-aware data selection.

- Authors introduce a Bidirectional Object-Water Attention (BiOW-Attn) module that enhances realism by modeling object-water interactions under multi-condition inputs.

- To select data samples that are most beneficial for detection performance, authors propose an Attribute-dependent Active Sampling (AAS) strategy that estimates training difficulty across semantic dimensions and selects high-value samples through difficulty-aware weighting.

- Authors also construct a Maritime Generation Dataset (MGD) for maritime detection, by pooling together 11,900 images from various publicly available and newly captured datasets – with corresponding captions, water surface masks, and bounding box annotations. The dataset covers five object categories, three viewpoints, four locations, and six imaging environments.

- Table 2 demonstrate that Neptune-X is better generator compared to baselines, and Table 3 and Fig 5 demonstrate that training on generated samples leads to improved performance across the board.

**Questions:**

- I wonder if the authors have ablations on ATDF factors to evaluate their individual contribution to the AAS strategy?

- Can the authors provide examples of generation diversity — for instance, using the same layout with different seeds or text prompts, and vice versa? This would help assess how well the model supports diverse outputs.

- Is the model capable of editing existing real images (e.g., inserting new objects into real maritime scenes), and if so, could this be used to further improve detection performance?

**Ethical Concerns:**

["NO or VERY MINOR ethics concerns only"]

**Final Justification:**

I thank the authors for their explanations and detailed experiments on saturating performance and ATDF contributions. I will raise my score in light of this.

Ideally, it would have been nice to verify the generated visuals - but I understand that the conference prevents visuals in rebuttal phase, which is a broader issue.

**Limitations:**

Yes.

**Paper Formatting Concerns:**

No.

**Quality:**

3

**Strengths And Weaknesses:**

Strengths:

- This work proposes a novel technique specifically tailored to improving methods for maritime detection, which represents a valuable contribution to the field.

- The thorough experiments and results clearly demonstrate the effectiveness of both the generator (Neptune-X) and the post-processing components (AAS strategy and ATDF factors).

- The authors have collected and annotated a dedicated dataset, the Maritime Generation Dataset (MGD), which could be beneficial for the broader research community.

- The paper is well-written, includes clear illustrative figures, and provides sufficient details about the experiments and the collected dataset.

Weaknesses:
- The modeling contribution, particularly the Bidirectional Object-Water Attention (BiOW-Attn) module, while effective, lacks strong novelty.

- The AAS strategy shows clear benefits in the low-data regime (~5K samples), but its marginal gains at 10K suggest diminishing returns. A more detailed explanation for this saturation would be helpful.

---

> ### Author Rebuttal · Authors · 2025-07-29
>
> We sincerely appreciate Reviewer tKB5's valuable feedback. Our responses to the weaknesses and questions are listed below.
>
> **R-W1 Effective but not strongly novel** We sincerely thank the reviewer for recognizing the effectiveness of our approach. While we acknowledge that the BiOW-Attn module builds upon existing attention mechanisms, our contributions extend beyond this single component. Our work makes three major contributions: (1) the BiOW-Attn mechanism, which is tailored to capture bidirectional object-water interactions, a critical yet underexplored factor in maritime scenes; (2) the novel AAS strategy that integrates active learning with attribute-aware sampling for synthetic data selection; and (3) the construction of the first benchmark dataset (MGD) specifically designed for maritime scene generation.
>
> More importantly, this represents the first attempt to systematically apply generative models to the maritime domain—an important but often overlooked application scenario. We hope this pioneering effort in addressing the unique challenges of maritime object detection through synthetic data generation will not be overlooked, rather than focusing solely on the novelty of individual module designs.
>
> **R-W2 Marginal gains of AAS** Thank you for your suggestion, which is similar to the **Reviewer S8Ax's W1.3** question. We conducted more extensive experiments to illustrate this saturation effect. As shown in the table below, there is a clear saturation effect when the data volume increases from 10k to 20k samples. This phenomenon primarily comes from our proposed AAS method, which pre-selects the most valuable generated samples. The samples ranked lower in the selection process contribute minimally to detection performance improvement since the model has already performed well on similar cases. This active selection strategy enables our method to achieve faster performance improvements with lower computational costs, reducing training overhead while maintaining effectiveness. To better illustrate this point, we will add this discussion in the final version and include 2D curve visualization to better demonstrate the relationship between sample quantity increase and detection performance improvement.
>
> | Generated Samples | mAP   | mAP@0.5 |
> |-------------------|-------|---------|
> | Baseline (0k)     | 39.99 |  61.13  |
> | 5k                | 43.11 |  64.70  |
> | 10k               | 43.62 |  65.50  |
> | 20k               | 43.63 |  65.70  |
> | 30k               | 43.82 |  65.92  |
> | 50k               | 44.06 |  66.05  |
>
> **R-Q1 Individual ATDF contribution** The table below demonstrates the individual contribution of each attribute dimension (ATDF) used in our AAS strategy for sample selection. As more attribute dimensions are incorporated, higher-value samples are prioritized, leading to progressive performance improvements. This validates the effectiveness of our AAS method in combining multi-dimensional attributes to select the most valuable synthetic samples for detection enhancement.
>
> | Viewpoint | Location | Imaging Environment | Object Category | mAP   | mAP@0.5 |
> |-----------|----------|---------------------|-----------------|-------|---------|
> | √         |          |                     |                 | 40.26 | 62.09   |
> | √         | √        |                     |                 | 41.27 | 62.26   |
> | √         | √        | √                   |                 | 41.40 | 63.10   |
> | √         | √        | √                   | √               | 43.62 | 65.50   |
>
> **R-Q2 More visualization of generation diversity** Due to the rebuttal policy restrictions, we are unable to provide additional anonymous visualizations during this phase. However, we will include comprehensive visual examples in the final version to illustrate the generation diversity of our model under the same layout with different seeds and text prompts.
>
> While these specific cases are not shown in the current version, we kindly refer the reviewer to Figures 8–11 in the appendix. These figures demonstrate the model’s ability to generate visually diverse maritime scenes without noticeable repetition in ship appearance, water texture, or overall scene composition, highlighting the model’s capacity for variation.
>
> **R-Q3 Image editing ability** Our model supports scene-level editing by modifying the input layout, which enables the generation of diverse maritime scenes. While direct editing of real images (e.g., inserting objects into real maritime scenes) is not the focus of our work, our generator, like other diffusion models, can be used in conjunction with existing image editing techniques to perform such tasks.
>
> Due to rebuttal policy restrictions, we cannot include additional visualizations at this stage, but we will provide relevant examples and discussion in the final version.

---

> > ### Comment · Reviewer_tKB5 · 2025-08-05
> >
> > I thank the authors for their explanations and detailed experiments on saturating performance and ATDF contributions. I will raise my score in light of this.
> >
> > Ideally, it would have been nice to verify the generated visuals - but I understand that the conference prevents visuals in rebuttal phase, which is a broader issue.

---

### Official Review · Reviewer_S8Ax · 2025-07-02

**Clarity:** 3
**Significance:** 2
**Originality:** 3
**Rating:** 5
**Confidence:** 5

**Summary:**

This paper studies the problem of using synthetically generated images to improve object detection in a new domain. It proposes a framework named Neptune-X to actively select and balance synthetically generated images for domain specific fine-tuning. To facilitate this, this paper introduces a conditional image generator called X-to-Maritime, building on top of existing Stable Diffusion [Ref 25] framework. Specifically, X-to-Maritime leverage several application-specific design including multi-conditional guidance, layout condition embedder, and bidirectional object-water attribution for improved control and quality. For end-to-end evaluation, the paper applies the Neptune-X to curate a large-scale synthetic dataset, which in turn boosts the YOLO object detection in the Maritime domain.

**Questions:**

Please address the comments especially [W1.1], [W1.2], [W1.3] and [W2] mentioned in the previous question.

**Ethical Concerns:**

["NO or VERY MINOR ethics concerns only"]

**Final Justification:**

Thanks for providing detailed response! Raised my rating from 4 to 5, assuming [W2] & [W3] will be addressed in the final rating.

**Quality:**

2

**Strengths And Weaknesses:**

**Strengths**

- [S1] This paper addresses a very practical domain adaptation problem. Specifically, it tackles both data sacristy and poor generation in maritime object detection, building on top of a foundational text-to-image diffusion model.

- [S2] While most of the work in the literature focused on generative model design, this paper presents an end-to-end solution to the problem, featured by the attribute-dependent active sampling technique.

**Weaknesses**

- [W1] The current evaluation does not clearly indicate  the specific impact of the proposed X-to-Maritime image synthesis engine on the final object detection performance. While Table 3 demonstrates performance gains when augmenting training data with generated samples, it remains ambiguous how much of this improvement is attributable to the enhanced synthesis capabilities of X-to-Maritime itself, as opposed to the overall data augmentation strategy.
  - [W1.1] To provide a more comprehensive understanding of the X-to-Maritime engine's contribution, it would be beneficial to include a comparative analysis of YOLO detector performance when retrained on synthetic data generated by existing state-of-the-art text-to-image synthesis methods, such as SD1.5 [Ref 25], RC-L2I [Ref 5], InstDiff [Ref 33], GLIGEN [Ref 16] , and LayoutDiff [Ref 42]. This would clearly highlight the direct impact of X-to-Maritime's generated data quality on object detection improvement.
  - [W1.2] Furthermore, extending the ablation studies (currently focused on generation quality in Table 4 ) to demonstrate the contribution of different components of the X-to-Maritime generation model (e.g., BiOW-Attn, multi-condition guidance) directly to the object detection improvement metrics (rather than YOLO score from a pre-trained detector) would offer deeper insights into the effectiveness of each generative module for downstream task performance.
  - [W1.3] To more comprehensively illustrate the impact of the proposed methods, it would be beneficial to present the results as a 2D curve, with mean Average Precision (mAP) on the Y-axis and the number of synthetic samples on the X-axis. This visualization would allow for a clearer understanding of the performance trends as the quantity of generated data increases. Specifically, it would be insightful to determine if adding, for example, 10x the current amount of synthetic data continues to yield improvements in object detection performance. Furthermore, such a curve would help elucidate the sustained importance of both the AAS strategy and the X-to-Maritime model in scenarios with significantly larger synthetic datasets.

- [W2] The paper states that the YOLO score was computed using an object detector pre-trained on synthetic data (L533-L537). This approach appears to be an unfair measurement when evaluating the quality of the synthetic data, as it would make more sense to use other object detectors trained on a larger, independent dataset to provide auxiliary signals.

- [W3] Figure 2 depicts the proposed Embedder taking separate Layout and Water Surface images, suggesting distinct processing streams. Clarification is needed regarding whether the embedder employs a separate auto-encoder for encoding Layout and Water Surface information in the actual implementation. If this is the case, it raises concerns about the generalizability of the approach to other related domains. Such a design might imply a significant reliance on domain-specific expertise or extensive pre-training tailored to maritime scenes, potentially limiting its direct applicability or requiring substantial adaptation for different environments or object types.

---

> ### Author Rebuttal · Authors · 2025-07-29
>
> We sincerely appreciate Reviewer S8Ax's valuable feedback. Our responses to the weaknesses and questions are listed below.
>
> **R-W1.1 Extensive comparison of generative models for detection improvement** Thank you for your suggestion. To demonstrate the benefits of our X-to-Maritime image generator in object detection, we conducted comparisons with state-of-the-art methods. The performance improvements of these methods on YOLOv10 detector are shown in the table below. It is clear that our method achieves the best performance in improving maritime detector performance, primarily attributed to the design of the BiOW-Attn module, which significantly enhances the modeling and control capabilities between objects and water bodies, thereby generating more realistic images. We will add further discussion in the final version.
>
> | Methods       | mAP   | mAP@0.5 |
> |---------------|-------|---------|
> | w/o           | 39.99 | 61.13   |
> | LayoutDiff    | 40.03 | 61.01   |
> | GLIGEN        | 41.54 | 62.85   |
> | InstDiff      | 41.32 | 62.57   |
> | RC-L2I        | 41.48 | 63.26   |
> | Neptune-X     | 43.62 | 65.50   |
>
> **R-W1.2 Extensive comparison of different configurations for detection improvement** Thank you for your suggestion. The benefits of each component to object detection improvement are shown in the table below. It is evident that using only simple object conditions can hardly effectively improve performance, mainly because it tends to produce poor generation results (e.g., ships separated from water surface). Simply adding water conditions can further enhance detection performance, but the improvement is still limited. However, introducing the bidirectional attention module for object-water modeling can significantly enhance the realism of synthetic data, thereby better improving downstream detector performance. We will add this experiment in the final version.
>
> | ObjCA | WatCA | Obj2WatCA | Wat2ObjCA | mAP     | mAP@0.5 |
> |-------|-------|-----------|-----------|---------|---------|
> | w/o   | w/o   | w/o       | w/o       | 39.99   | 61.13   |
> | √     |       |           |           | 40.94   | 61.18   |
> | √     | √     |           |           | 41.09   | 61.15   |
> | √     | √     | √         |           | 42.54   | 63.85   |
> | √     | √     |           | √         | 42.75   | 63.93   |
> | √     | √     | √         | √         | 43.62   | 65.50   |
>
> **R-W1.3 Correlation between detection accuracy and number of synthetic samples** Due to the rebuttal requirment constraints, we present the results in tabular format. We conducted data augmentation experiments on YOLOv10 using 5k, 10k, 20k, 30k, and 50k synthetic samples, respectively. The experimental results are shown in the table below.
>
> | Generated Samples | mAP   | mAP@0.5  |
> |-------------------|-------|----------|
> | Baseline (0k)     | 39.99 |  61.13   |
> | 5k                | 43.11 |  64.70   |
> | 10k               | 43.62 |  65.50   |
> | 20k               | 43.63 |  65.70   |
> | 30k               | 43.82 |  65.92   |
> | 50k               | 44.06 |  66.05   |
>
> It is worth noting that there is a clear saturation effect when the data volume increases from 10k to 20k samples. This phenomenon primarily stems from our proposed AAS method, which pre-selects the most valuable generated samples. The samples ranked lower in the selection process contribute minimally to detection performance improvement since the model has already performed well on similar cases. This active selection strategy enables our method to achieve faster performance improvements with lower computational costs, reducing additional training overhead while maintaining effectiveness. We will include the suggested 2D curve visualization in the final version to more intuitively demonstrate this.
>
> **R-W2 YOLO score** We thank the reviewer for this concern, which may stem from a misunderstanding. In fact, the detector used for YOLO score computation is pre-trained on large-scale datasets and then fine-tuned on our constructed small-scale maritime dataset (which does not contain any synthetic data). The fine-tuning is necessary because our dataset includes objects that are rare in general scenarios but particularly important in maritime contexts, such as buoys. The detector has never used any data generated by generative models during the training, so there is no unfairness issue. We hope this clarification addresses your concern.
>
> **R-W3 Water condition embedding** This concern may stem from a misunderstanding caused by Figure 2. Actually, we use a single embedder to encode both object and water surface conditions simultaneously, rather than employing separate auto-encoders. We will modify Figure 2 in the final version to avoid this unclarity.

---

### Official Review · Reviewer_EyhV · 2025-07-04

**Clarity:** 4
**Significance:** 4
**Originality:** 4
**Rating:** 5
**Confidence:** 5

**Summary:**

This paper introduces Neptune-X, a framework for improving maritime object detection by generating synthetic training data with a generative model and selecting the most useful synthetic samples via an active sampling strategy. The authors highlight that maritime datasets suffer from scarce, costly annotation and poor diversity across key attributes (object category, viewpoint, environment, etc.).

Neptune-X addresses this in two main ways:

1. X-to-Maritime Generation: A diffusion-based generator (built on Stable Diffusion) extended with a Bidirectional Object-Water Attention (BiOW-Attn) module to model interactions between objects and the water surface. This improves the realism and semantic coherence of synthetic maritime scenes under multi-modal conditions.

2. Attribute-dependent Active Sampling (AAS): An approach that scores generated samples by how difficult they are for an object detector, using Attribute-correlated Training Difficulty Factors (ATDFs) across dimensions like category, viewpoint, location, and environment, to prioritize hard or underrepresented cases.

They also create the Maritime Generation Dataset (MGD), the first large-scale dataset designed for training generative maritime models, with over 11,000 images and rich annotations.

Extensive experiments show improvements in synthetic image quality (measured via FID, CAS, and YOLO Score) and object detection accuracy (via mAP) when augmenting detectors with selected synthetic data. An ablation study demonstrates the benefit of BiOW-Attn and AAS over baselines.

**Questions:**

1. BiOW-Attn: Could the authors provides qualitative result of the data generated without water condition? I am not fully motivated the purpose of water condition. Can I understand the water condition is just like the semantic mask condition telling the model here is the water? Can we just use traditional semantic mask to do this instead of designing a new attention layer?

2. Sampling Strategy Comparison: How does the Attribute-dependent Active Sampling compare to standard active learning heuristics (e.g., entropy sampling, core-set)? Could the authors include such baselines?

3. Detector Dependence: The pre-trained detector used for ATDF scoring is critical. How sensitive is the overall framework to the quality of this detector? Would a worse initial detector degrade AAS selection?

Broader Impact Discussion: The paper briefly notes no negative societal impacts, but generated imagery could plausibly be misused for deception (e.g., fake maritime surveillance images). Would the authors consider adding even a short discussion acknowledging this risk?

**Ethical Concerns:**

["NO or VERY MINOR ethics concerns only"]

**Final Justification:**

In the rebuttal the authors provide new experiments result addressing my concern on the Yolo detector and the generalizability of the method.

I also continuously appreciate the strengths of the paper:

1. Clear Motivation & Importance: The maritime domain has real and specific data scarcity issues that make synthetic generation important and meaningful. I like the way how the authors adjust existing techniques in ML community to support the domain-specific applications.

2. Technical Contributions: The BiOW-Attn module is a well-motivated addition, specifically modeling object–water boundary interactions for more realistic generation. The AAS sampling framework integrates active learning principles into generative augmentation, addressing the problem that not all synthetic data is equally useful.

3. Strong Empirical Evaluation: Multiple metrics (FID, CAS, YOLO Score) on generation. Detection performance improvements (7–10% mAP gains) on multiple YOLO baselines. Ablation studies isolating the effect of attention components and sampling strategies.

4. Dataset Contribution: The new Maritime Generation Dataset (MGD) is well-documented, diverse in viewpoints, locations, and conditions, with an explicit plan for release.

Therefore I remains my score and recommend acceptance.

**Limitations:**

The authors do include a limitations discussion acknowledging reliance on discrete attribute categories and proposing more continuous or hierarchical modeling in future work. They don't discuss possible misuse (e.g., generating fake surveillance imagery). I recommend adding that. Overall they are upfront about technical limitations but could improve societal impact reflection.

**Paper Formatting Concerns:**

None found. The paper appears to follow NeurIPS guidelines correctly.

**Quality:**

4

**Strengths And Weaknesses:**

## Strengths

1. Clear Motivation & Importance: The maritime domain has real and specific data scarcity issues that make synthetic generation important and meaningful. I like the way how the authors adjust existing techniques in ML community to support the domain-specific applications.

2. Technical Contributions: The BiOW-Attn module is a well-motivated addition, specifically modeling object–water boundary interactions for more realistic generation. The AAS sampling framework integrates active learning principles into generative augmentation, addressing the problem that not all synthetic data is equally useful.

3. Strong Empirical Evaluation: Multiple metrics (FID, CAS, YOLO Score) on generation. Detection performance improvements (7–10% mAP gains) on multiple YOLO baselines. Ablation studies isolating the effect of attention components and sampling strategies.

4. Dataset Contribution: The new Maritime Generation Dataset (MGD) is well-documented, diverse in viewpoints, locations, and conditions, with an explicit plan for release.

## Weaknesses


1. Evaluation Scope: The object detection experiments only use YOLO variants; it’s unclear if improvements generalize to other detection architectures. For example, can grounding dino be used as a object detector to evaluate the dataset?

2. Limited Baseline Comparisons in Sampling:
While random sampling is compared to AAS, other active learning sampling strategies (e.g., uncertainty-based, diversity-based) are not evaluated. Also, ATDF is introduced with some mathematical sophistication, but the benefits over simpler scoring are only superficially explored.

3. Assumptions and Generalization:
The ATDF method relies on predefined discrete attributes (category, viewpoint, etc.) which the authors admit limits granularity. But there’s no experiment showing how much that matters.
The approach is tightly tailored to maritime scenes; it's unclear if BiOW-Attn or AAS generalize to other domains with different context interactions---However, I don't think this point affect the value of this paper as we already claimed we are focusing on the maritime domain.

---

> ### Author Rebuttal · Authors · 2025-07-29
>
> We sincerely appreciate Reviewer EyhV's valuable feedback. Our responses to the weaknesses, questions, and limitations are listed below.
>
> **R-W1 Limited to YOLO variants** Our proposed method is indeed applicable to various types of detectors, including open-vocabulary detectors such as Grounding DINO. To demonstrate this generalization, we conducted additional experiments using Grounding DINO as the object detector. As shown in the table below, our proposed data generation method significantly enhances Grounding DINO's detection performance, achieving substantial improvements in mAP and mAP@0.5. This demonstrates that the effectiveness of our generation approach extends beyond common detectors represented by the YOLO variants to open-vocabulary detectors.
>
> | Methods                   | mAP            | mAP@0.5        |
> |---------------------------|----------------|----------------|
> | Grounding DINO (official) | 8.42           | 12.60          |
> | Grounding DINO (fintuned) | 65.03          | 86.12          |
> | + Gen Data                | 68.04 (+4.63%) | 89.86 (+4.34%) |
>
> **R-W2 & Q2 Sampling strategy comparison** Thank you for your insightful comment. To fully demonstrate the advantages of our proposed method, we conducted comprehensive comparisons with 5 different active learning approaches, including uncertainty-based sampling methods (Entropy, Variance, and Margin) and diversity-based sampling methods (Greedy K-Center and K-Means Corset). For diversity-based sampling, we constructed 7-dimensional features using detection box count, average box area, box area standard deviation, average confidence, confidence standard deviation, average box x-coordinate, average box y-coordinate, and category count.
>
> | Sampling Methods | mAP   | mAP@0.5 |
> |------------------|-------|---------|
> | w/o              | 39.99 | 61.13   |
> | Entropy          | 42.62 | 64.40   |
> | Variance         | 42.42 | 64.17   |
> | Margin           | 42.87 | 64.55   |
> | Greedy K-Center  | 42.24 | 63.90   |
> | K-Means Corset   | 42.27 | 63.79   |
> | AAS              | 43.62 | 65.50   |
>
> The results show that while these traditional methods also improve performance, our proposed AAS method achieves superior results by more comprehensively considering maritime-specific attributes such as water conditions, weather, and viewpoint information, leading to better detection performance.
>
> **R-W3 Assumptions and generalization** We thank the reviewer for the thoughtful comment. While our method is specifically designed for maritime scene generation and detection, certain components may offer insights or inspiration for other domains that face similar challenges. For instance, the generation module could be adapted to tasks requiring fine-grained control over multi-condition interactions, and the AAS sampling strategy may be relevant to object detection problems involving multi-dimensional attribute distributions. We will expand the discussion in the final version to clarify the potential for such cross-domain relevance and the associated limitations.
>
> **R-Q1 BiOW-Attn and water condition** The purpose of BiOW-Attn is not merely to introduce water conditions, but to better capture the interaction relationships between water bodies and water surface objects, thereby improving generation quality. The qualitative visualization results (Figure 10) and quantitative metrics (Table 2) comparing scenarios without water conditions and with simple water condition addition are already provided in our paper. From the experimental analysis, approaches that ignore or simply introduce water conditions tend to produce issues such as objects floating in the air above water and unrealistic water-object interaction boundaries. Our designed BiOW-Attn effectively addresses these problems by modeling the bidirectional attention between objects and water surfaces, leading to improved generation quality.
>
> **R-Q3 ATDF's detector dependence** The purpose of ATDF is to observe the detection performance differences of specified detector across different attribute dimensions, thereby enabling the targeted selection of generated samples that better improve corresponding detector performance. Therefore, a worse initial detector does not degrade the effectiveness of AAS selection. This has been experimentally validated in Table 3 of the main text, where three detectors with different performance levels were used. When fine-tuned with the samples selected based on their respective ATDFs, all detectors showed significant performance improvements, demonstrating the robustness of our AAS strategy across different detector qualities.
>
> **R-Q4 & R-L Broader impact discussion** While our work is intended primarily for maritime safety, monitoring, and research applications, we acknowledge that synthetic image generation capabilities could potentially be misused for creating deceptive maritime surveillance content. We will add a broader impact discussion in the final version that addresses these concerns and emphasizes the importance of responsible deployment of our framework.

---

> > ### Comment · Reviewer_EyhV · 2025-08-04
> >
> > Thanks for the rebuttal! Most of my concerns are addressed.

---

### Decision · Program_Chairs · 2025-09-17

**Decision:**

Accept (spotlight)

**Comment:**

The paper received four expert reviews. The authors provided a rebuttal that attempted to address the concerns raised in the reviews. The reviewers read the rebuttal and engaged with the authors.  The reviewers unanimously like the paper and recommended accept. The area chair agreed with the recommendation and decided to accept the paper. Congratulations! Please see the reviews for feedback on the paper to revise the final version of your paper and include any items promised in your rebuttal.